# FAM210A is essential for cold-induced mitochondrial remodeling in brown adipocytes

Jiamin Qiu[1,7], Feng Yue [1,2,7] ✉, Peipei Zhu[3], Jingjuan Chen[1], Fan Xu[4], Lijia Zhang[1], Kun Ho Kim [1], Madigan M. Snyder [1,5], Nanjian Luo[1], Hao-wei Xu[1], Fang Huang[4], W. Andy Tao [3,6] & Shihuan Kuang [1,6] ✉

Cold stimulation dynamically remodels mitochondria in brown adipose tissue (BAT) to facilitate non-shivering thermogenesis in mammals, but what regulates mitochondrial plasticity is poorly understood. Comparing mitochondrial proteomes in response to cold revealed FAM210A as a cold-inducible mitochondrial inner membrane protein. An adipocyte-specific constitutive knockout of *Fam210a* (*Fam210a^AKO*) disrupts mitochondrial cristae structure and diminishes the thermogenic activity of BAT, rendering the *Fam210a^AKO* mice vulnerable to lethal hypothermia under acute cold exposure. Induced knockout of *Fam210a* in adult adipocytes (*Fam210a^iAKO*) does not affect steady-state mitochondrial structure under thermoneutrality, but impairs cold-induced mitochondrial remodeling, leading to progressive loss of cristae and reduction of mitochondrial density. Proteomics reveals an association between FAM210A and OPA1, whose cleavage governs cristae dynamics and mitochondrial remodeling. Mechanistically, FAM210A interacts with mitochondrial protease YME1L and modulates its activity toward OMA1 and OPA1 cleavage. These data establish FAM210A as a key regulator of mitochondrial cristae remodeling in BAT and shed light on the mechanism underlying mitochondrial plasticity in response to cold.

Mitochondria are multifaceted metabolic organelles that alter their density and architecture dynamically to meet diverse metabolic demands[1]. These changes are mainly mediated by mitochondrial fusion and fission, as well as biogenesis and remodeling of mitochondrial cristae. Mitochondrial cristae morphology is critical for mitochondrial function, as it affects the assembly of respiratory chain complexes and supercomplexes[2,3]. Emerging studies have demonstrated that mitochondrial cristae undergo dynamic changes in response to metabolic demands and physiological stimuli[4–6]. Cristae dynamics are regulated by mitochondrial inner membrane remodeling machinery, including dynamin-related GTPase optic atrophy type 1 (OPA1)[7] and mitochondrial contact site and cristae organizing system

(MICOS)[8,9]. As such, OPA1 and MICOS have been shown to be required for cellular adaption to energy substrate availability[10,11].

Brown and beige adipocytes are mitochondria-enriched cells capable of dissipating energy in the form of heat[12]. Brown adipocyte thermogenesis is potently activated by cold, which stimulates the sympathetic nervous system that innervates the brown adipocytes. Norepinephrine (NE) released from sympathetic nerves activates β3-adrenergic receptors (β3-ARs) on brown adipocytes, triggering a signaling cascade that facilitates the uptake of energy substrates (such as fatty acids (FAs), glucose, and amino acids) to promote cellular metabolism, predominantly mitochondrial metabolism[13,14]. FAs also activate uncoupling protein 1 (UCP1), a FA/H$^+$ symporter located on the

---

[1]Department of Animal Sciences, Purdue University, West Lafayette, IN 47907, USA. [2]Department of Animal Sciences, University of Florida, Gainesville, FL 32611, USA. [3]Department of Biochemistry, Purdue University, West Lafayette, IN 47907, USA. [4]Weldon School of Biomedical Engineering, Purdue University, West Lafayette, IN 47907, USA. [5]Department of Biological Sciences, Purdue University, West Lafayette, IN 47907, USA. [6]Purdue University Institute for Cancer Research, West Lafayette, IN 47907, USA. [7]These authors contributed equally: Jiamin Qiu, Feng Yue. ✉e-mail: fengyue@ufl.edu; skuang@purdue.edu

mitochondrial inner membrane, to dissipate the H+ gradient and generate heat[15,16]. Besides acting as a thermogenic organ, recent advances have highlighted the biological significance of brown adipose tissue (BAT) in systemic energy homeostasis and endocrine function, including its role in regulating glucose uptake, lipid metabolism, insulin sensitivity, and global adipose tissue homeostasis[17–19].

Recent studies have implicated the importance of mitochondrial plasticity in regulating the thermogenic function of brown and beige adipocytes[20,21]. Upon cold exposure, mitochondria undergo dynamic remodeling in morphology and function to fulfill the energy demand for maintaining body temperature. Conditional knockout (KO) of *Opa1* causes fragmented mitochondria and disrupted cristae structure, which leads to the conversion of brown adipocytes (BAs) into cells resembling white adipocytes[22]. Interestingly, overexpression of *Opa1* converts white adipocytes into beige adipocytes, thereby promoting browning of white adipose tissue (WAT) and improving systemic glucose tolerance and insulin sensitivity[23]. Moreover, OMA1-mediated OPA1 processing is essential for fully activation of BAT thermogenesis, and loss of OMA1 results in obesity and defective thermogenesis[24,25]. Taken together, these studies underscore the crucial role of mitochondrial cristae of adipocytes in regulating systemic metabolism. Nevertheless, the molecular mechanisms mediating cristae remodeling in response to metabolic cues in fat tissues remain elusive.

In the present study, we surveyed the mitochondrial proteomes at various time points during cold-induced thermogenesis and cristae remodeling in BAT of mice. This approach led to the identification of FAM210A (Family with sequence similarity 210 member A) as a cold-induced mitochondrial inner membrane protein. Deletion of *Fam210a* in BAs impairs mitochondrial cristae remodeling and eventually leads to the loss of mitochondrial cristae, the whitening of BAT, and the failure of cold-induced thermogenesis in mice. Using a cell-free protein expression system combined with biochemical and pharmacological inhibition assays in vitro and in vivo, we further demonstrate that FAM210A governs OPA1 cleavage by modulating YME1L/OMA1 activity to facilitate mitochondrial cristae dynamics under cold challenge. These results uncover a regulatory axis underlying mitochondrial plasticity and thermogenesis in BAT.

## Results

### Cold-induced mitochondrial cristae remodeling and proteomic changes in BAT

To systematically explore the mitochondrial dynamics in BAT, wild-type C57BL/6J mice housed at thermoneutrality (TN, 30 °C) were challenged with cold at 6 °C, and mitochondrial cristae morphologies were examined by using transmission electron microscopy (TEM) at various time points before and after the cold treatment (Fig. 1a). As expected, mitochondrial cristae underwent dynamic remodeling in response to changes in ambient temperature[26]. Specifically, mitochondria with loose and irregularly arranged cristae (named as Class 1) were primarily found in BAT of mice under TN (Fig. 1a). After 6–24 h of cold exposure, a majority of mitochondria displayed reduced and curved cristae (Class 2) which were arranged in two or more concentric semicircles (Fig. 1a). After 3 d of cold exposure, cristae density was dramatically increased, and the morphology of cristae became straight and lamellar (Class 3), which were more closely spaced and tightly packed at 7 d (Fig. 1a). Quantitatively, the percentage of Class 1 mitochondria was decreased sharply after cold exposure, Class 2 mitochondria percentage peaked at 6 h and 24 h, while the percentage of Class 3 mitochondria was markedly increased at 3 d and 7 d after cold exposure (Fig. 1b).

To understand the molecular underpinning of cristae remodeling, we performed an unbiased BAT mitochondrial proteomic analysis at five different time points before and after cold exposure (Fig. 1c). A total of 998 proteins were consistently detected in all five groups (Supplementary Fig. 1a). Principle component analysis showed that the proteomes were clearly clustered by time points (Supplementary Fig. 1b). Heatmap clustering further revealed highly consistent cold-induced expression signatures in mitochondrial proteomes (Fig. 1d and Supplementary Data 1). Compared to the TN proteome, the number of cold-induced proteins was increased with cold exposure, as 71, 179, 272, and 462 proteins significantly changed at 6 h, 24 h, 3 d, and 7 d, respectively (Supplementary Fig. 1c and Supplementary Data 2). As an indication of the reliability of our proteomic data, several known markers of BAT activation were gradually induced in cold-exposed mice (Supplementary Fig. 1d). Gene ontology (GO) cellular component analysis revealed that the differentially expressed proteins (DEPs) were enriched in the mitochondrial inner membrane (Fig. 1e and Supplementary Fig. 1e), in line with the observed cristae remodeling (Fig. 1a, b). GO biological process analysis of DEPs showed that a large number of proteins were enriched in oxidation-reduction processes and transport (Supplementary Fig. 1f). Using the mitochondrial protein functional classifications of MitoCoP[27], we found that a majority of mitochondrial inner membrane proteins had higher expression levels at 3 d and 7 d, relative to their TN expression levels (Supplementary Fig. 1g–i).

### Identification of FAM210A as a cold-induced trans-inner membrane mitochondrial protein

To identify candidate proteins involved in regulating cold-induced mitochondrial dynamics, we specifically focused on proteins related to cristae formation, mitochondrion organization, and mitochondrial fusion as defined by GO biological processes (Fig. 1f). This narrowed the list down to 15 proteins, including OPA1 and MICOS proteins, which directly control mitochondrial cristae structures[8,28], as well as YME1L, a protease responsible for OPA1 cleavage[29,30]. These proteins were significantly changed in their expression levels following cold exposure (Fig. 1g). The immunoblotting analysis (Fig. 1h and Supplementary Fig. 1j) highlighted a pronounced change in OPA1 cleavage patterns. Under TN condition, the long form of OPA1 (L-OPA1) was predominately detected, but its expression significantly reduced after 6 h and 24 h of cold exposure, accompanied by a marked increase of the short form of OPA1 (S-OPA1) (Fig. 1h). Over the subsequent 3–7 d of cold exposure, the expression of L-OPA1 increased while S-OPA1 decreased (Fig. 1h). These dynamic changes in L/S-OPA1 ratio suggest active regulation of OPA1 cleavage during BAT thermogenesis. Echoing with the cristae ultrastructural changes (Fig. 1a, b), these results demonstrate an association between the temporal dynamics of OPA1 processing and cristae remodeling during BAT thermogenesis.

To explore the uncharacterized regulators of cristae remodeling, we further analyzed the mitochondrial proteomic data and found that FAM210A, a predicted mitochondrial protein containing domain of unknown function (DUF) (Fig. 2a), was rapidly induced by cold at 6 h and highly expressed at 7 d (Fig. 2b). Immunoblotting analysis confirmed the upregulation of FAM210A during thermogenesis (Fig. 2c). As predicted, FAM210A contains a mitochondrial targeting signal (MTS) at its N-terminus, followed by a transmembrane domain, a DUF1279 domain, and a coiled-coil domain at its C-terminus (Fig. 2d). RoseTTAfold predicted that the DUF1279 domain is in a helix structure (Fig. 2d). FAM210A was highly expressed in mature BAs (Fig. 2e), and immunostaining analysis showed that FAM210A colocalized with MitoTracker in both brown preadipocytes (0 d) and differentiated BAs (8 d) (Fig. 2f). Single-molecule localization microscopy confirmed the mitochondrial localization of adenovirus-transduced *Fam210a* in Cos-7 cells and revealed its non-overlapping localization with the outer membrane marker TOM20 (Fig. 2g). A protease protection assay further revealed that FAM210A is a trans-inner membrane protein with the N-terminus in the mitochondrial intermembrane space and a ~12-kDa C-terminus in the mitochondrial matrix (Fig. 2h). This C-terminus was protected from protease digestion after swelling (Fig. 2h).

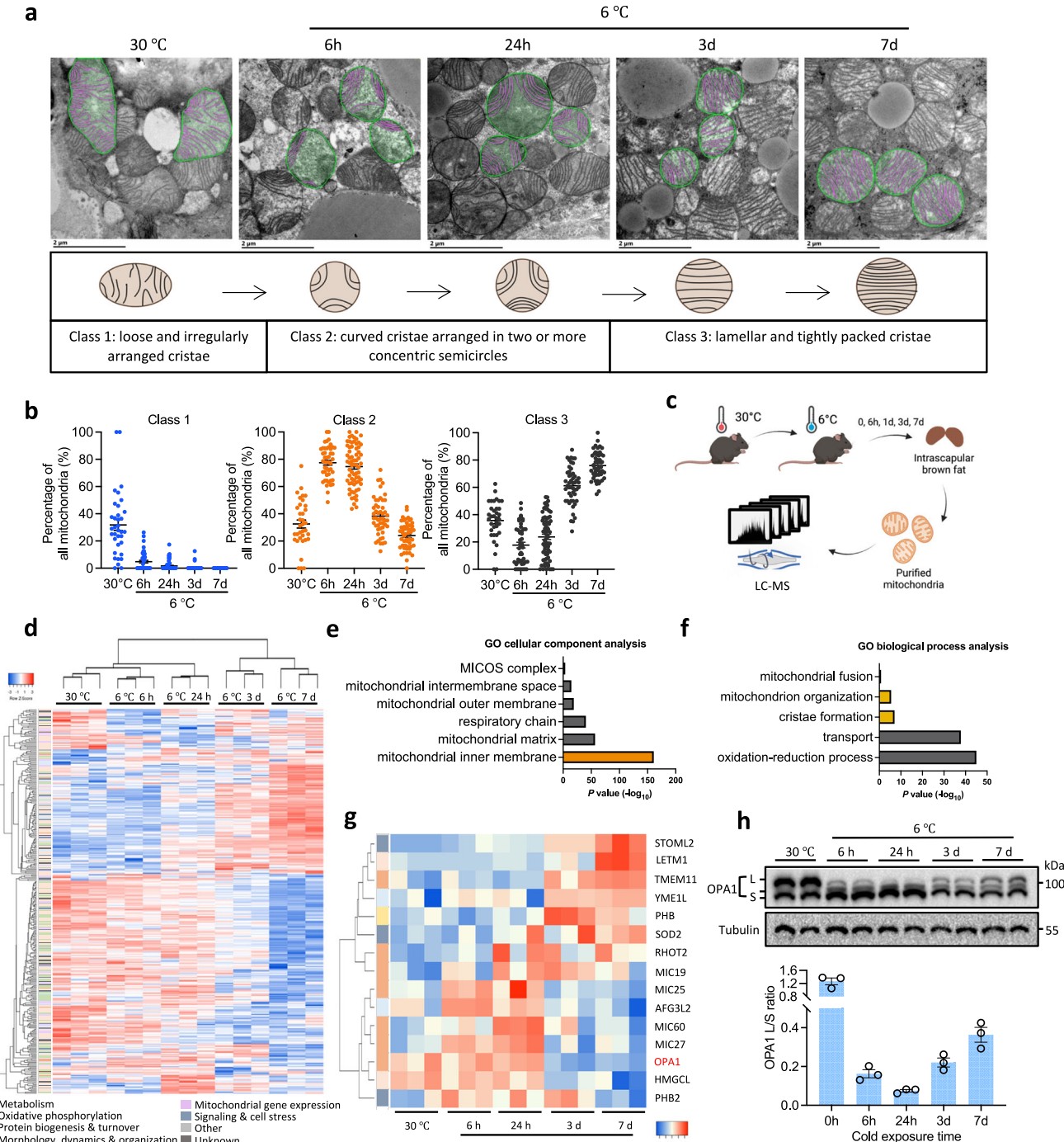

**Fig. 1 | BAT mitochondrial cristae remodeling and associated changes in proteomes during cold-induced thermogenesis. a** Representative transmission electron microscopy (TEM) images showing mitochondrial remodeling in the wild-type brown adipose tissue (BAT) upon cold exposure for different times (*n* = 3 mice per group; scale bars, 2 μm; diagram created with BioRender.com). **b** Quantification of mitochondrial morphological changes in (**a**) (*n* = 35 images for 30 °C; *n* = 47 images for 6 h on 6 °C; *n* = 47 images for 6 h on 6 °C; *n* = 65 images for 24 h on 6 °C; *n* = 50 images for 3 d and 7 d on 6 °C; 3 mice were used at each time point; number of mitochondria counted in 30 °C, and 6 h, 24 h, 3 d, 7 d on 6 °C: 539, 909, 2133, 1047, 1041; mean ± s.e.m). **c** Experimental workflow for the study of BAT mitochondrial proteomics of mice upon cold exposure for different times (LC-MS, liquid chromatography-mass spectrometry; diagram created with BioRender.com). **d** Heatmap of differentially expressed mitochondrial protein expression level in BAT upon cold exposure for different times (*n* = 3 mice per group; one-way ANOVA; differential expression was determined using a cutoff significance level of *P* < 0.05). **e** Gene Ontology (GO) cellular component analysis of differentially expressed proteins (Fisher's Exact test; *P* values were adjusted by Benjamini−Hochberg method). **f** GO biological process analysis of differentially expressed proteins from the subset of mitochondrial inner membrane in (**e**) (Fisher's Exact test; *P* values were adjusted by Benjamini−Hochberg method). **g** Heatmap of differentially expressed proteins of cristae formation, mitochondrion organization, and mitochondrial fusion. **h** Immunoblotting and quantification analysis of OPA1 (*n* = 3 mice per group; mean ± s.e.m). Source data are provided as a Source Data file.

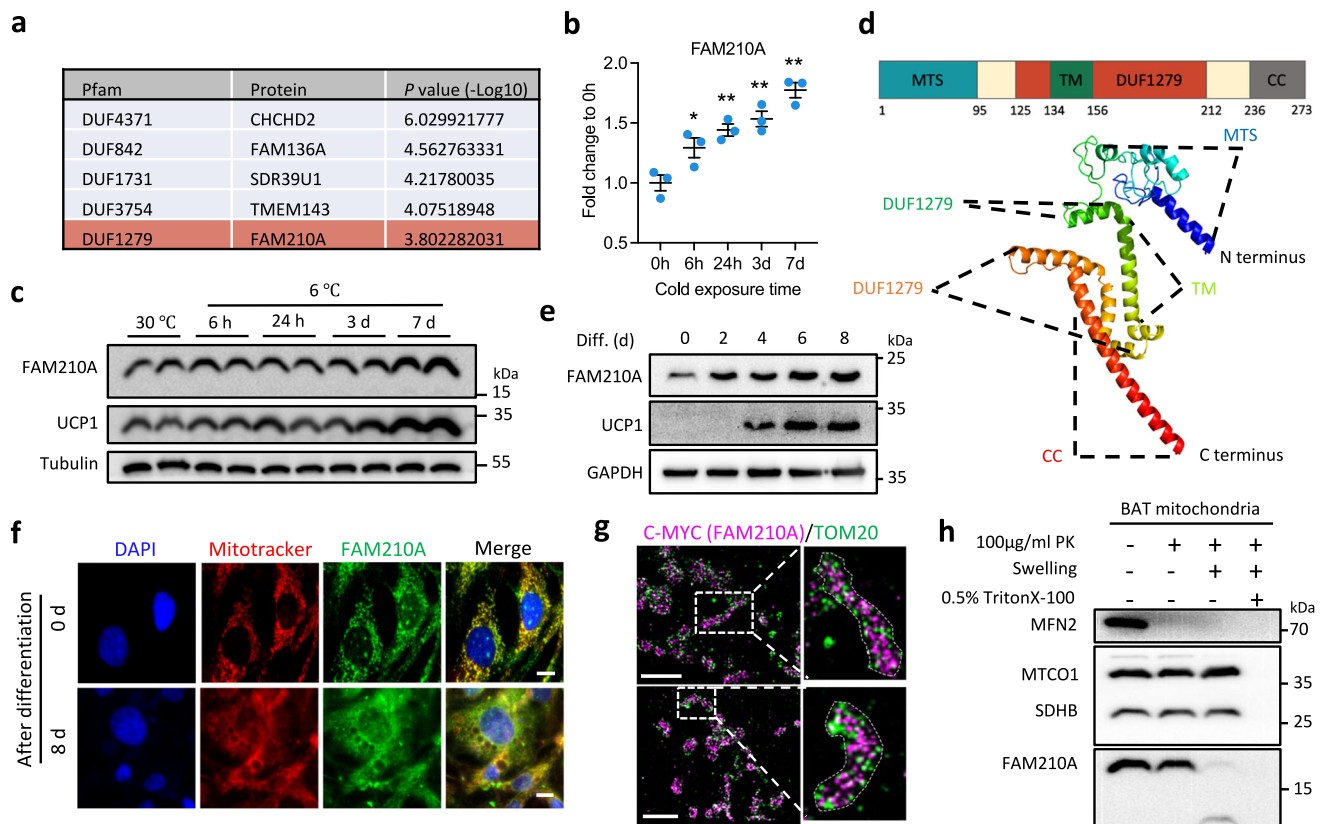

**Fig. 2 | Identification of FAM210A as a cold-inducible mitochondrial inner membrane protein in BAT. a** The list of differentially expressed mitochondrial proteins with domain of unknown function (DUF) (One-way ANOVA). **b** Quantification of FAM210A expression in mitochondrial proteomic data ($n = 3$ mice per group; mean ± s.e.m; two-tailed unpaired Student's $t$ test; $P = 0.0497$, 0.006, 0.0044, and 0.0011). **c** Immunoblotting analysis of FAM210A in BAT from mice upon cold exposure at different time points ($n = 3$ mice per group). **d** Schematic showing the predicated domain structure and RoseTTAfold prediction of FAM210A. MTS, mitochondrial targeting sequence; TM, transmembrane region; DUF1279, domain of unknown function 1279; CC, coiled-coil domain. **e** Immunoblotting analysis of FAM210A in brown adipocytes (BAs) during differentiation ($n = 3$ independent experiments). **f** Immunofluorescence showing the colocalization of FAM210A with mitochondria in BAs before and after differentiation ($n = 3$ independent experiments; scale bars, 10 μm). **g** Single-molecule localization image of overexpressed *Fam210a* in Cos-7 cells ($n = 3$ independent experiments; scale bars, 2 μm). **h** Protease protection assay with mitochondrial fractions purified from BAT mitochondria (PK, proteinase K; $n = 3$ independent experiments). *$P < 0.05$, **$P < 0.01$. Source data are provided as a Source Data file.

## Loss of *Fam210a* causes whitening of BAT

To understand the physiological function of FAM210A, we generated adipocyte-specific *Fam210a* KO mice (*Fam210a^{AKO}*) by crossing *Fam210a^{f/f}* mice to *Adiponectin-Cre* (*Adipoq-Cre*) transgenic mice (Fig. 3a). *Fam210a^{AKO}* mice were born normally and had comparable body weight and tissue mass (Supplementary Fig. 2a–c), but *Fam210a^{AKO}* BAT exhibited obvious whitening compared to control mice (Fig. 3b). Histological analysis revealed that while the inguinal white adipose tissues (iWAT) were morphologically similar in control and *Fam210a^{AKO}* mice, classic multilocular BAs found in the interscapular BAT (iBAT) of control mice were replaced by unilocular white adipocyte-like cells in the *Fam210a^{AKO}* BAT (Fig. 3c). Single cell analysis confirmed the multilocular lipid droplet composition of adipocytes isolated from control BAT but the unilocular morphology of adipocytes isolated from *Fam210a^{AKO}* mice (Fig. 3d). As the BAT in *Fam210a^{AKO}* mice developed normally in newborns (Supplementary Fig. 2d–f), the whitening that occurred postnatally is most likely due to cold stress after birth.

We also examined whether the whitening of BAT in *Fam210a^{AKO}* mice leads to metabolic dysfunction. Control and *Fam210a^{AKO}* mice were subjected to an indirect calorimetry measurement to evaluate their energy expenditure and metabolic activity at room temperature (RT, 22 °C). We did not observe any significant difference in energy expenditure, $O_2$ consumption, and $CO_2$ production between genotypes (Supplementary Fig. 2g–j). We further performed

intraperitoneal glucose tolerance test (GTT) and insulin tolerance test (ITT) to assess the glucose clearance rate and insulin sensitivity in *Fam210a^{AKO}* mice. We found no change in GTT but did observe significantly blunted ITT in *Fam210a^{AKO}* mice (Supplementary Fig. 2k, l). Moreover, we measured the serum lipid profiles and found comparable levels of non-esterified fatty acids (NEFAs), cholesterol, high-density lipoproteins (HDLs), low-density lipoproteins (LDLs), and triglycerides between genotypes (Supplementary Fig. 2m). Together, these results suggest that genetic deletion of *Fam210a* in adipocytes impairs BAT functions and systemic insulin sensitivity.

## FAM210A is required for cold-induced thermogenesis

We next investigated the thermogenic function of *Fam210a^{AKO}* mice during acute and chronic cold exposure. In response to acute cold exposure in the absence of food, significantly lower rectal temperatures were observed in *Fam210a^{AKO}* mice, which unanimously exhibited hypothermia (<34 °C) within 3 h (Fig. 3e, f). In response to chronic cold exposure in the presence of food, the *Fam210a^{AKO}* mice displayed similar rectal temperatures to control mice after 7 d of cold exposure (Supplementary Fig. 2n). However, infrared thermography and temperature microprobe both detected a significantly lower temperature in the iBAT of *Fam210a^{AKO}* mice than in control mice (Fig. 3g–i). These results indicate that the BAT of *Fam210a^{AKO}* mice is defective in cold-induced thermogenesis.

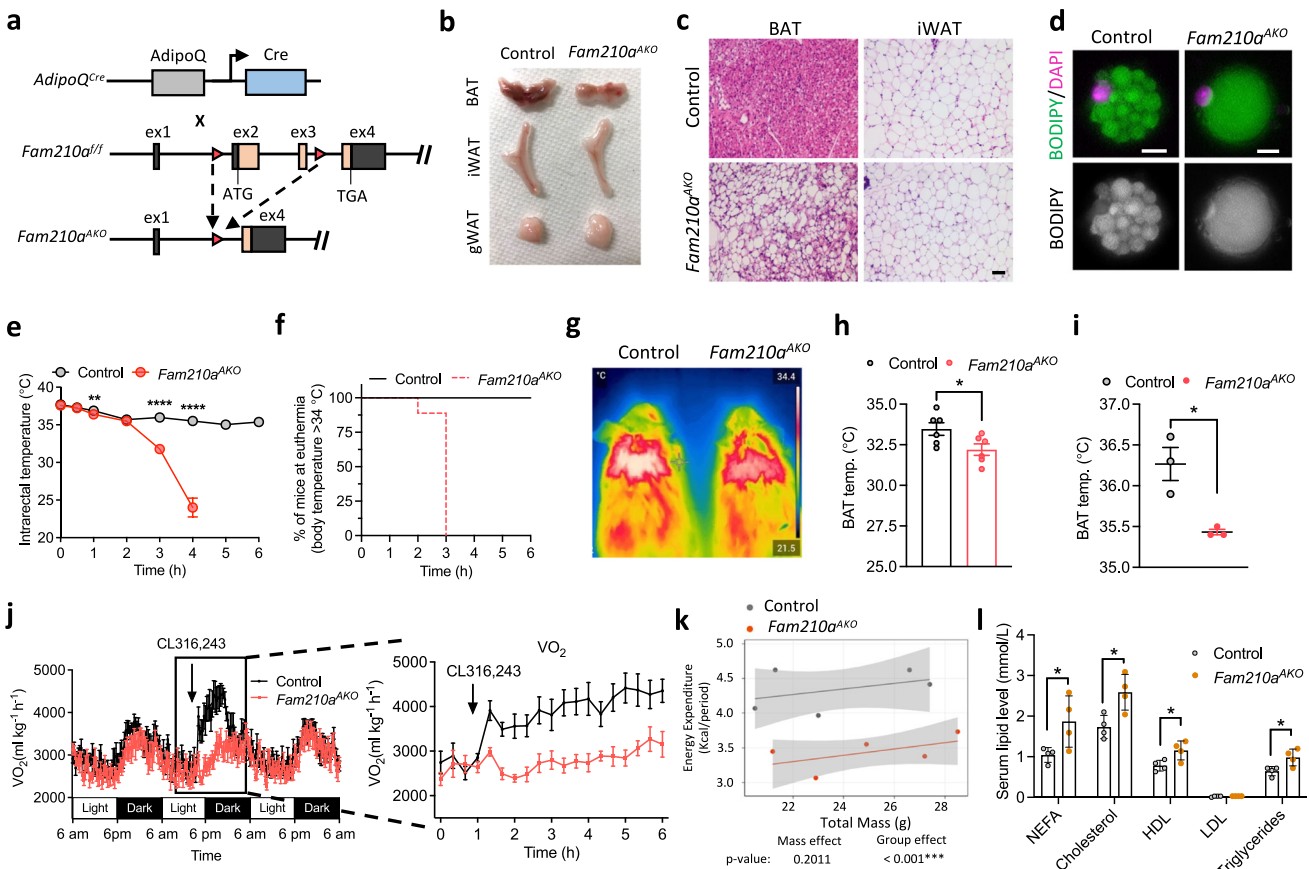

**Fig. 3 | Loss of *Fam210a* in adipocytes causes whitening of BAT and cold intolerance in mice. a** Schematic diagram of the strategy used to generate adipocyte-specific *Fam210a* knockout (*Fam210a^{AKO}*) mice (ex, exon). **b** Representative image showing the morphology of adipose tissues from control and *Fam210a^{AKO}* mice at room temperature (RT, 22 °C) (iWAT, inguinal white adipose tissue; gWAT, gonadal white adipose tissue; *n* = 4 mice per group). **c** Representative H&E staining of BAT and iWAT from control and *Fam210a^{AKO}* mice at RT (*n* = 4 mice per group; scale bar, 50 μm). **d** BODIPY staining of mature BAs isolated from control and *Fam210a^{AKO}* mice at RT (*n* = 4 mice per group; scale bars: 10 μm). **e** Rectal core body temperature of control and *Fam210a^{AKO}* mice following acute cold exposure with fasting (control, *n* = 10 mice; *Fam210a^{AKO}*, *n* = 9 mice; mean ± s.e.m; two-tailed unpaired Student's *t* test; *P* = 0.039056, 0.00000089, and 0.000000036). **f** Survival curves of control and *Fam210a^{AKO}* mice following acute cold exposure with fasting (euthermia is higher than 34 °C) (Control, *n* = 10 mice; *Fam210a^{AKO}*, *n* = 9 mice). **g** Infrared thermography of iBAT temperature in control and *Fam210a^{AKO}* mice upon cold exposure for 7 d (*n* = 6 mice per group).

**h** Quantification of BAT temperature in (**g**) (mean ± s.e.m; two-tailed unpaired Student's *t* test; *P* = 0.0367). **i** iBAT temperature of control and *Fam210a^{AKO}* mice upon cold exposure for 7 d by using temperature microprobe (*n* = 3 mice per group; mean ± s.e.m; two-tailed unpaired Student's *t* test; *P* = 0.0154). **j** Oxygen consumption at light and dark. The left panel in (**j**) shows the oxygen consumption for 2 d; the right panel in (**j**) highlights the oxygen consumption after CL316,243 injection (*n* = 5 mice per group; mean ± s.e.m). **k** Regression-based analysis of energy expenditure in (**j**) against body mass. Data were analyzed using CalR-ANCOVA with energy expenditure as a dependent variable and body mass as a covariate. The *P* values (ANCOVA) are shown at the bottom (*n* = 5 mice per group; mean ± s.e.m). **l** Serum lipid profiles in control and *Fam210a^{AKO}* mice after cold exposure for 7 d (NEFA non-esterified fatty acid, HDL high-density lipoprotein, LDL low-density lipoprotein; *n* = 4 mice per group; mean ± s.e.m; two-tailed unpaired Student's *t* test; *P* = 0.04608, 0.017339, 0.030646, and 0.028823). *$P < 0.05$, **$P < 0.01$, ****$P < 0.0001$. Source data are provided as a Source Data file.

Cold exposure induces BAT thermogenesis through β3-ARs, which can be mimicked by the β3-AR agonist, CL316,243 (CL)[31]. Using the indirect calorimetry approach, we examined the metabolic responses of control and *Fam210a^{AKO}* mice to CL-stimulation at RT. While CL treatment induced a significant increase in O₂ consumption in control mice, the *Fam210a^{AKO}* mice failed to respond to this stimulation (Fig. 3j, k). In addition, upon cold exposure, *Fam210a^{AKO}* mice exhibited lower blood glucose levels (Supplementary Fig. 3a) and higher levels of serum lipids (NEFAs, cholesterol, HDLs, and triglycerides) compared to control mice (Fig. 3l), indicating a systematic change in glucose and lipid metabolism. These results together suggest that FAM210A is indispensable for cold-induced adipose tissue thermogenesis and metabolic adaptation.

### Loss of *Fam210a* impairs mitochondrial function and compromises cold-induced browning

We also examined responses of adipose tissues to chronic cold exposure, which induces browning of WAT in mice. Tissue masses of both iWAT and gonadal WAT (gWAT) were higher in *Fam210a^{AKO}* mice (Fig. 4a, b). Cold-exposed *Fam210a^{AKO}* mice had pale BAT filled with cells containing unilocular lipid droplets resembling white adipocytes (Fig. 4a, c). The iWATs of *Fam210a^{AKO}* mice were also whiter than those of control mice after chronic cold exposure (Fig. 4a). H&E and immunohistochemistry staining revealed the formation of beige adipocytes with multilocular lipid droplets and UCP1 expression in iWAT of control mice, but not *Fam210a^{AKO}* mice (Fig. 4c, d). The mRNA levels of browning markers *Ucp1*, *Ppargc1a*, and *Dio2* were significantly decreased in both BAT and iWAT of *Fam210a^{AKO}* mice compared with control mice (Fig. 4e). Consistently, the protein levels of UCP1 and OXPHOS complexes (except for complex V protein ATP5A) were dramatically reduced (Fig. 4f–i). These data suggest that FAM210A is required for cold-induced adipose tissue browning.

This finding led us to examine if mitochondrial function is deficient in the absence of *Fam210a*. Indeed, *Fam210a^{AKO}* BAT had diminished fatty acid oxidation (FAO) activity (Fig. 4j). We also

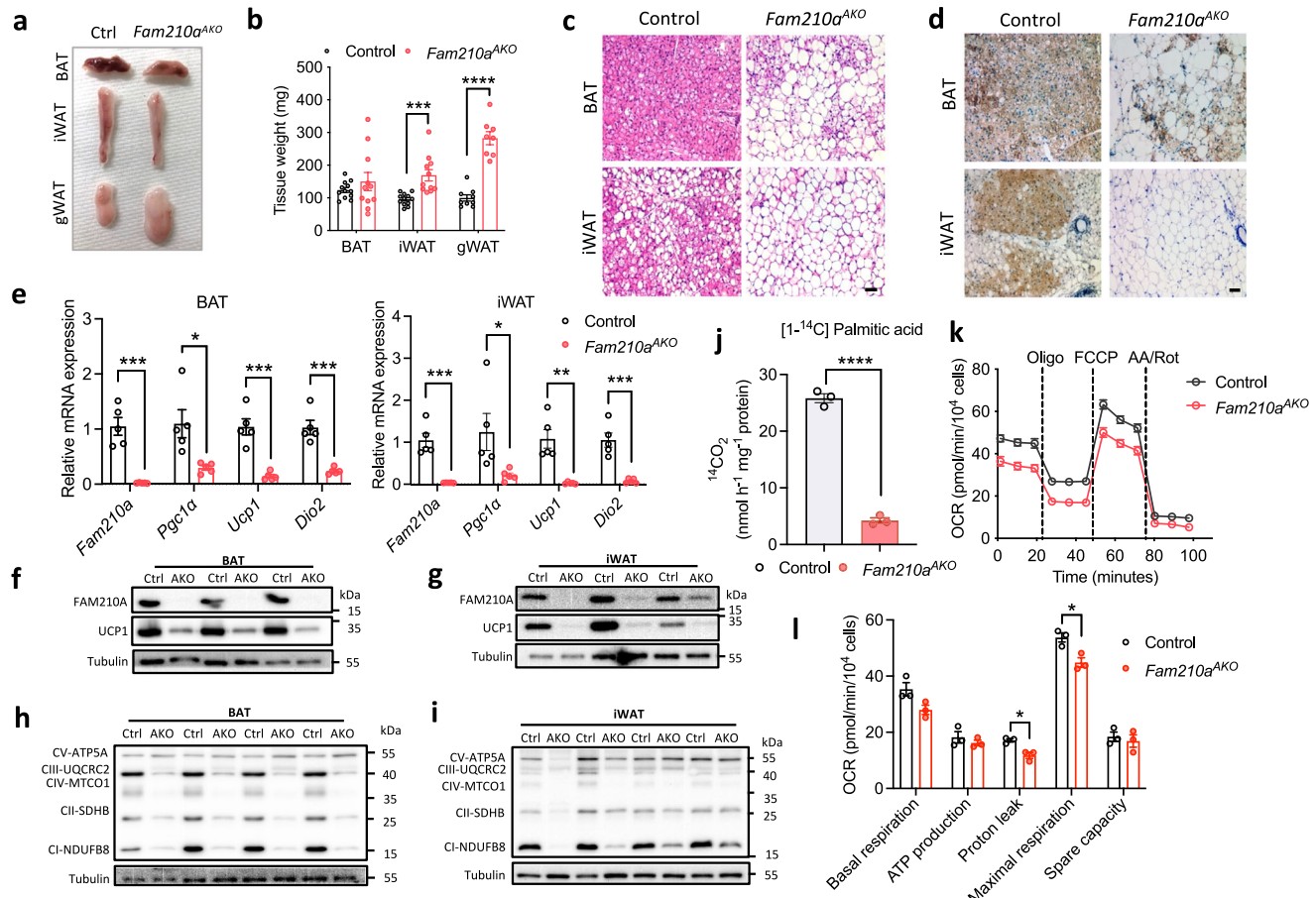

**Fig. 4 | Loss of *Fam210a* impairs cold-induced browning of adipocytes and mitochondrial function. a, b** Representative images showing the morphology (**a**) and tissue mass (**b**) of adipose tissues from control and *Fam210a^AKO* mice after cold exposure for 7 d (*n* = 12 mice for control BAT and iWAT; *n* = 9 mice for control gWAT; *n* = 11 mice for *Fam210a^AKO* BAT and iWAT; *n* = 8 mice for *Fam210a^AKO* gWAT; mean ± s.e.m; two-tailed unpaired Student's *t* test; *P* = 0.000388, and <0.000001). **c** Representative H&E staining of BAT and iWAT from control and *Fam210a^AKO* mice after cold exposure for 7 d (*n* = 4 mice per group; scale bar, 50 μm). **d** Representative immunohistochemical staining of UCP1 in BAT and iWAT from control and *Fam210a^AKO* mice after cold exposure for 7 d (*n* = 3 mice per group; Scale bar: 50 μm). **e** mRNA level of thermogenic genes in BAT and iWAT of control and *Fam210a^AKO* mice after cold exposure for 7 d (*n* = 5 mice per group; mean ± s.e.m; two-tailed unpaired Student's *t* test; *P* = 0.000207, 0.0145, 0.000309,

0.000328, 0.00029, 0.049247, 0.001473, and 0.000464). **f, g** Immunoblotting analysis of UCP1 in BAT (**f**) and iWAT (**g**) from control (Ctrl) and *Fam210a^AKO* (AKO) mice (*n* = 3 mice per group). **h, i** Immunoblotting analysis of BAT (**h**) and iWAT (**i**) mitochondrial proteins in control and *Fam210a^AKO* mice after cold exposure (*n* = 4 mice per group). **j** Fatty acid oxidation (FAO) activity measured using ^14C-labeled palmitic acid in BAT from control and *Fam210a^AKO* mice with cold exposure for 7 d (*n* = 3 mice per group; mean ± s.e.m; two-tailed unpaired Student's *t* test; *P* = 0.0000198). **k** The oxygen consumption rate (OCR) of differentiated primary BAs from control and *Fam210a^AKO* with CL316,243 treatment for 6 h before the test (Oligo Oligomycin, AA Antimycin A, Rot Rotenone; *n* = 3 mice per group; mean ± s.e.m). **l** Quantification of OCR from (**k**) (mean ± s.e.m; two-tailed unpaired Student's *t* test; *P* = 0.011873 and 0.019458). **P* < 0.05, ***P* < 0.01, ****P* < 0.001, *****P* < 0.0001. Source data are provided as a Source Data file.

measured oxygen consumption rate (OCR) to determine the coupled and uncoupled respiration of BAs differentiated from stromal vascular fraction (SVF) cells that were isolated from BAT of control and *Fam210a^AKO* mice (Fig. 4k). The *Fam210a^AKO* BAs exhibited lower UCP1-dependent and maximal (combined UCP1-dependent and UCP1-independent) respiration (Fig. 4l). In contrast, the OCR for ATP production was similar in control and *Fam210a^AKO* cells (Fig. 4l). These data suggest that the deletion of *Fam210a* results in impaired mitochondrial function and FAO in BAT.

Reduced mitochondrial FAO is often compensated by glucose metabolism. Consistent with this notion, the blood glucose level was lower in *Fam210a^AKO* than control mice after cold (Supplementary Fig. 3a). We also tracked the glucose uptake of different tissues after cold exposure and detected a very specific increase in glucose uptake in BAT of *Fam210a^AKO* mice compared to control mice, associated with elevated GLUT4 levels in *Fam210a^AKO* mice (Supplementary Fig. 3b–d). Despite the increased glucose uptake, *Fam210a^AKO* BAs showed a similar glycolytic rate as control BAs (Supplementary Fig. 3e, f). Taken

together, these findings imply that *Fam210a* ablation causes mitochondrial dysfunction, reduced FAO, and compensatory increased glucose uptake in BAT.

## FAM210A regulates cold-induced cristae and mitochondrial remodeling in BAT

To directly explore how the loss of *Fam210a* affects mitochondria, we first examined mitochondrial abundance. Significantly lower amounts of mitochondria were isolated from *Fam210a^AKO* BAT than control BAT (Fig. 5a). Along this line, fewer mitochondrial DNA (mtDNA) copy numbers were detected in *Fam210a^AKO* BAT compared to control BAT (Fig. 5b, c). In iWAT, however, reduced mtDNA copy numbers were only observed in the *Fam210a^AKO* mice after chronic cold exposure, but not at RT (Fig. 5b, c). As chronic cold exposure promotes the recruitment of mitochondria-enriched beige adipocytes in WAT[32], our finding suggests that *Fam210a^AKO* impairs cold-induced mitochondria biogenesis. To further visualize the morphological change of mitochondria, we performed TEM and found that the size of mitochondria in

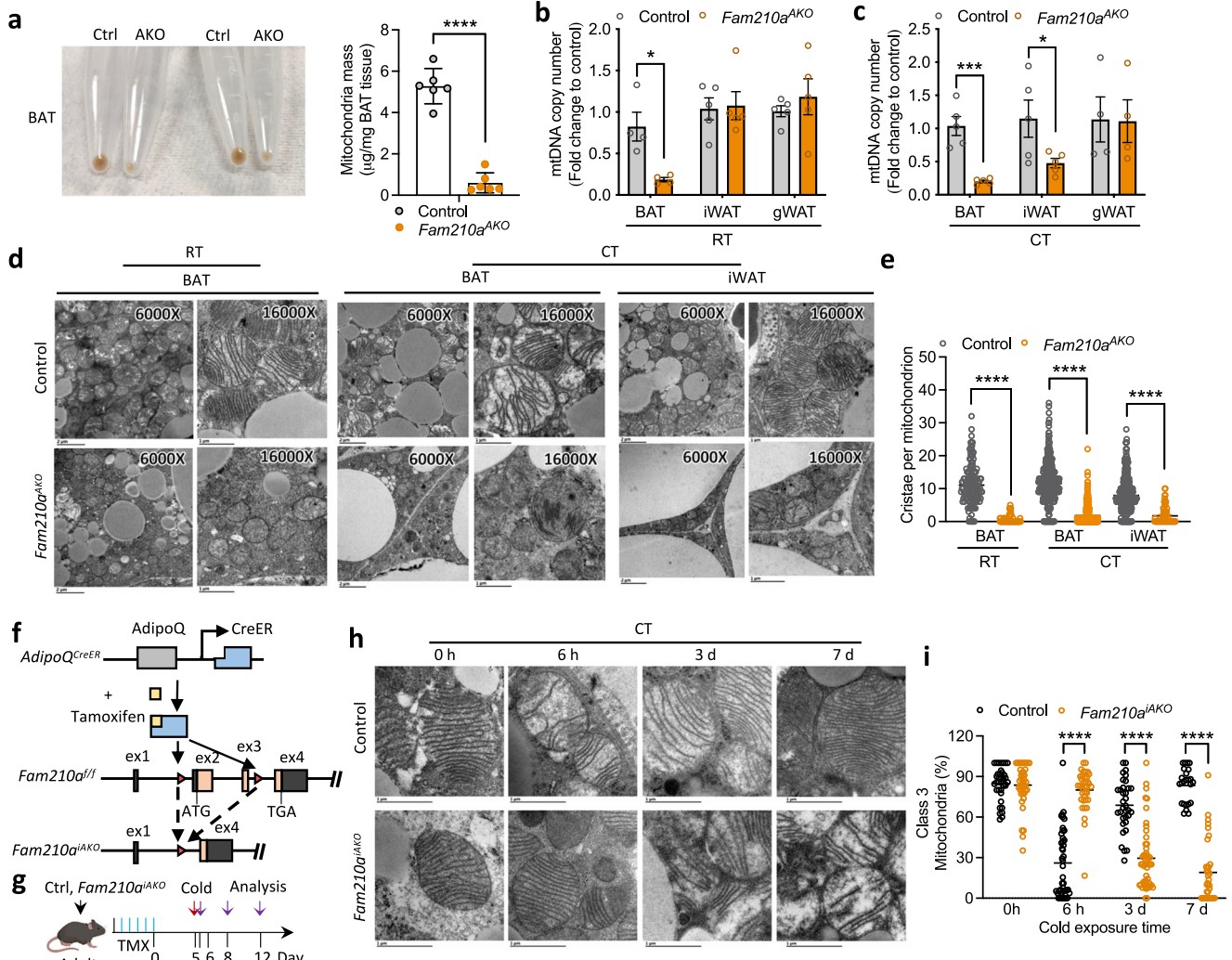

**Fig. 5 | Fam210a is required for cristae and mitochondrial remodeling in BAT.**
**a** Representative images and quantification of isolated mitochondria pellets from control (Ctrl) and *Fam210a^AKO* (AKO) BAT (*n* = 6 mice per group; mean ± s.e.m; two-tailed unpaired Student's *t* test; *P* = 0.0000003729). **b** Mitochondria copy number in BAT, iWAT, and gWAT from control and *Fam210a^AKO* mice at room temperature (RT) (*n* = 4 mice for control BAT; *n* = 5 mice for control iWAT and gWAT; *n* = 4 mice for *Fam210a^AKO*; mean ± s.e.m; two-tailed unpaired Student's *t* test; *P* = 0.010882). **c** Mitochondria copy number in BAT, iWAT, and gWAT from control and *Fam210a^AKO* mice upon cold exposure (CT) for 7 d (*n* = 5 mice per group for BAT and iWAT; *n* = 4 mice per group for gWAT; mean ± s.e.m; two-tailed Student's *t* test; *P* = 0.000395 and 0.049103). **d** Representative TEM images of BAT and iWAT from control and *Fam210a^AKO* mice at RT and cold exposure for 7 d (*n* = 3 mice per group; for 6000X, scale bars, 2 μm; for 16000X, scale bars, 1 μm). **e** Quantification of cristae number per mitochondrion shown in (**d**) (*n* = the number of mitochondria counted in the order they appear on the graph from left to right: 156, 104, 431, 487, 459, 76; 3

mice were used for each group; mean ± s.e.m; two-tailed unpaired Student's *t* test; ****P* < 0.00000001). **f** Schematic diagram of the strategy used to generate inducible adipocyte-specific *Fam210a* knockout mice (*Fam210a^iAKO*). **g** Schematic showing timing of tamoxifen (TMX) induction, cold exposure, and sampling for control and *Fam210a^iAKO* mice (diagram created with BioRender.com).
**h** Representative TEM images of control and *Fam210a^iAKO* BAT upon cold exposure for different times (*n* = 3 mice per group; scale bars, 1 μm). **i** Quantification of Class 3 mitochondria in (**h**) (*n* = 37, 42, 34, and 24 images for the control group at 0 h, 6 h, 3 d, and 7 d after cold, respectively; *n* = 41, 38, 49, and 34 images for *Fam210a^iAKO* group at 0 h, 6 h, 3 d, and 7 d after cold, respectively; the number of mitochondria counted in the order they appear on the graph from left to right: 510, 620, 781, 559, 685, 861, 526, 521; 3 mice were used for each group; mean ± s.e.m; two-tailed unpaired Student's *t* test; ****P* < 0.00000001). **P* < 0.05, ***P* < 0.01, ****P* < 0.001, *****P* < 0.0001. Source data are provided as a Source Data file.

---

*Fam210a^AKO* BAT was much smaller than those in control BAT. Moreover, while control mitochondria showed lamellar cristae, *Fam210a^AKO* mitochondria, to our surprise, exhibited remarkably rare cristae (Fig. 5d, e), indicating an essential role of FAM210A in mitochondrial cristae integrity.

To exclude the effect of *Fam210a* deletion on mitochondria during BAT development, we generated inducible adipocyte-specific *Fam210a* KO mice (*Fam210a^iAKO*) by crossing *Fam210a^f/f* mice to the *Adipoq-CreER* mice (Fig. 5f). Five consecutive daily tamoxifen (TMX) injections were administered to induce *Fam210a* KO in adult mice (Supplementary Fig. 4a). We confirmed that TMX efficiently reduced FAM210A protein levels at d 5 after injection (Supplementary Fig. 4b).

BAT morphology and mitochondrial protein levels were indistinguishable in control and *Fam210a^iAKO* mice 3 weeks after TMX induction (Supplementary Fig. 4c, d). To further examine a potential role of FAM210A in BAT mitochondria integrity independent of cold-induced remodeling, the mice were housed at TN for two weeks after TMX induction (Supplementary Fig. 4e). Control and *Fam210a^iAKO* mice showed no difference in BAT and BAT mitochondria (Supplementary Fig. 4f–k). These results suggest that FAM210A is not essential for BAT mitochondrial maintenance and integrity under non-thermogenic condition.

We then exposed TMX-treated control and *Fam210a^iAKO* mice to 6 °C for 6 h to 7 d to investigate the role of FAM210A during cold-

induced cristae remodeling (Fig. 5g). Strikingly, TMX-treated *Fam210a*[iAKO] BAT mitochondria exhibited compromised cristae remodeling in response to cold challenge (Fig. 5h). Specifically, upon cold exposure, BAT mitochondria in control mice underwent a ~50% decrease in Class 3 mitochondria at 6 h, followed by a ~50% increase at 3 d. Additionally, cristae abundance steadily increased from 3 d to 7 d (Fig. 5h, i). By contrast, mitochondrial morphology in TMX-treated *Fam210a*[iAKO] BAT displayed no difference from 0 h to 6 h upon exposure to cold, while Class 3 mitochondria were decreased by ~85% from 6 h to 7 d, accompanied by a substantial loss of cristae (Fig. 5h, i). These data illustrate the crucial role of FAM210A in regulating cold-induced mitochondrial cristae remodeling in brown adipocytes.

The deficient cold-induced mitochondrial cristae remodeling was associated with functional defects, as the TMX-treated *Fam210a*[iAKO] BAT accumulated more lipid droplets and displayed lower tissue density than control mice at 7 d after cold exposure (Supplementary Fig. 5a, b). *Fam210a*[iAKO] BAT also had a lower FAO activity (Supplementary Fig. 5c). Moreover, TMX-treated *Fam210a*[iAKO] mice exhibited a significantly lower BAT temperature than control mice after 7 d cold exposure, while no differences were observed before or during the cold exposure (Supplementary Fig. 5d). Consistently, indirect calorimetry measurements showed that, while the CL-stimulated $O_2$ consumption and energy expenditure were comparable between *Fam210a*[iAKO] and control mice before cold exposure, lower levels of $O_2$ consumption and energy expenditure were observed in *Fam210a*[iAKO] mice after 7 d cold exposure (Supplementary Fig. 5e–h). These results demonstrate an indispensable role of FAM210A in BAT thermogenesis and sympathetic response under chronic cold challenge. The observation that *Fam210a* KO impaired mitochondrial cristae remodeling within 6 h of cold exposure without affecting BAT thermogenesis suggests that FAM210A mediates additional effects to influence the thermogenesis of BAT under chronic exposure.

### *Fam210a* deletion leads to excessive OPA1 cleavage in response to cold

To understand how FAM210A regulates cold-induced mitochondrial cristae remodeling, we performed mitochondrial proteomic analysis of control and *Fam210a*[iAKO] BAT. We exposed TMX-treated control and *Fam210a*[iAKO] mice to cold for 3 d and isolated mitochondria from BAT for proteomic analysis (Fig. 6a). This analysis confirmed marked reduction of FAM210A levels in *Fam210a*[iAKO] BAT (Supplementary Fig. 6a). A total of 855 proteins were detected in both control and *Fam210a*[iAKO] groups (Supplementary Fig. 6b), and 115 proteins exhibited significantly different levels between control and *Fam210a*[iAKO] mice (Supplementary Data 3). Notably, 85 of 115 differentially expressed proteins were downregulated in *Fam210a*[iAKO] BAT mitochondria (Fig. 6b and Supplementary Data 3). These proteins are associated with the GO terms "translation" and "transport" and are mainly located on "mitochondrial inner membrane" (Supplementary Fig. 6c). The results are consistent with our observation of decreased cristae abundance in *Fam210a*[iAKO] BAT.

A recent study reported that FAM210A influences pathological cardiac remodeling by regulating the translation of mitochondrial-encoded proteins[33]. To test whether *Fam210a* KO causes defects in mitochondrial protein translation in BAT, we examined the levels of selective nuclear genome- and mitochondrial genome-encoded mitochondrial localized proteins. Neither levels of nuclear encoded proteins (CI-NDUFB8, CII-SDHB, CIII-UQCRC, CV-ATP5A) nor levels of mitochondrial encoded proteins (CI-ND1, CIII-CYB, CIV-MTCO1, CIV-MTCO2, CV-ATP6) were altered by *Fam210a* KO at 6 h, 3 d, and 7 d after cold exposure (Supplementary Fig. 7a–d). These data suggest that *Fam210a* KO does not directly affect cold-induced mitochondrial protein translation in BAT.

We further screened proteins related to "mitochondrial morphology, dynamics, and organization"[27] and found that OPA1 was

the only protein whose level was significantly decreased by *Fam210a* KO (Fig. 6c). We validated the reduction of OPA1 levels (especially of L-OPA1) in *Fam210a*[AKO] and *Fam210a*[iAKO] BAT by immunoblotting (Fig. 6d–g and Supplementary Fig. 8a). Remarkably, the ratios of L-OPA1/S-OPA1 were significantly lower in *Fam210a*[iAKO] than in control BAT within 6 h of cold challenge, without changing the total OPA1 levels (Fig. 6f and Supplementary Fig. 8b). The accelerated and excessive cold-induced cleavage of OPA1 in the *Fam210a*[iAKO] BAT suggests that FAM210A normally protects OPA1 from excessive cleavage. Indeed, adenovirus-mediated overexpression of *Fam210a* (FamAd) in BAT elevated the ratios of L-OPA1/S-OPA1 in *Fam210a*[iAKO] at 24 h after cold exposure (Fig. 6h, i).

### FAM210A controls cold-induced OPA1 cleavage through YME1L/OMA1 axis

OPA1 cleavage is catalyzed by two peptidases, YME1L and OMA1[29,34]. To test if the excessive OPA1 cleavage in *Fam210a*[iAKO] BAT is due to the increased activity of YME1L and OMA1, we inhibited YME1L and OMA1 activity with a widely used inhibitor, o-phenanthroline (o-phe)[35–37], and found that both OPA1 cleavage and mitochondrial remodeling in *Fam210a*[iAKO] BAT were partially rescued by o-phe at 24 h after cold exposure (Fig. 6j–l). We also treated BAT explants with o-phe ex vivo and observed the rescue of OPA1 cleavage in the *Fam210a*[iAKO] BAT (Supplementary Fig. 8c, d). Together, these data indicate that the loss of *Fam210a* upregulates the activity of YME1L and OMA1, leading to excessive OPA1 cleavage that compromises cold-induced cristae remodeling in BAT (Fig. 6m).

To further dissect how FAM210A regulates YME1L and OMA1, we first examined mitochondrial membrane potential and ATP levels, as they have been shown to affect the activity of YME1L and OMA1[36,38,39]. BAT mitochondria isolated from control and *Fam210a*[iAKO] mice after 24 h of cold exposure showed similar membrane potential and ATP levels (Supplementary Fig. 9a–c), excluding their role in mediating FAM210A function. We next synthesized recombinant FAM210A, YME1L, OMA1, and OPA1 proteins and performed multiple cell-free assays. The recombinant proteins were stable in the reaction system (Supplementary Fig. 9d), and FAM210A had no effect on the stability of YME1L, OMA1, and OPA1 when they were individually incubated with FAM210A (Supplementary Fig. 9e–g). However, FAM210A upregulated the activity of YME1L without altering its level, based on the increased cleavage of OPA1 by YME1L in the presence of FAM210A (Fig. 7a, b and Supplementary Fig. 9h, i). In contrast, FAM210A did not affect the activity or level of OMA1, as OPA1 cleavage stayed the same with or without FAM210A (Fig. 7c, d and Supplementary Fig. 9j, k). Furthermore, co-immunoprecipitation (Co-IP) and proximity ligation assay (PLA) suggest the direct physical interaction between FAM210A and YME1L (Fig. 7e, f). As YME1L and OMA1 degrade each other reciprocally[1,36], we also examined if FAM210A plays a role in this process. We incubated OMA1 and YME1L together with or without FAM210A and observed that OMA1 degradation, but not YME1L degradation, was accelerated in the presence of FAM210A (Fig. 7g, h and Supplementary Fig. 9l, m). These results together suggest that FAM210A enhances YME1L's activity in OMA1 degradation and the KO of *Fam210a* should decrease YME1L's activity but increase OMA1's activity. To validate this, we performed the large-gel immunoblotting analysis of OPA1 to detect cleavage products preferentially cleaved by YME1L and OMA1[38,40,41]. This analysis revealed that *Fam210a* deletion enhanced OMA1 activity in BAT during cold exposure (Fig. 7i, j). In specific, *Fam210a*[iAKO] significantly decreased the ratio of OPA1-b to OPA1-e (Fig. 7i, j), a negative indicator of OMA1 mediated OPA1 cleavage[38,40–42]. Collectively, these findings establish a model in which FAM210A enhances YME1L activity, thereby restricting OMA1 and facilitating a specific cold-induced OPA1 cleavage pattern in BAT that favors cold-induced mitochondrial remodeling (Fig. 7k).

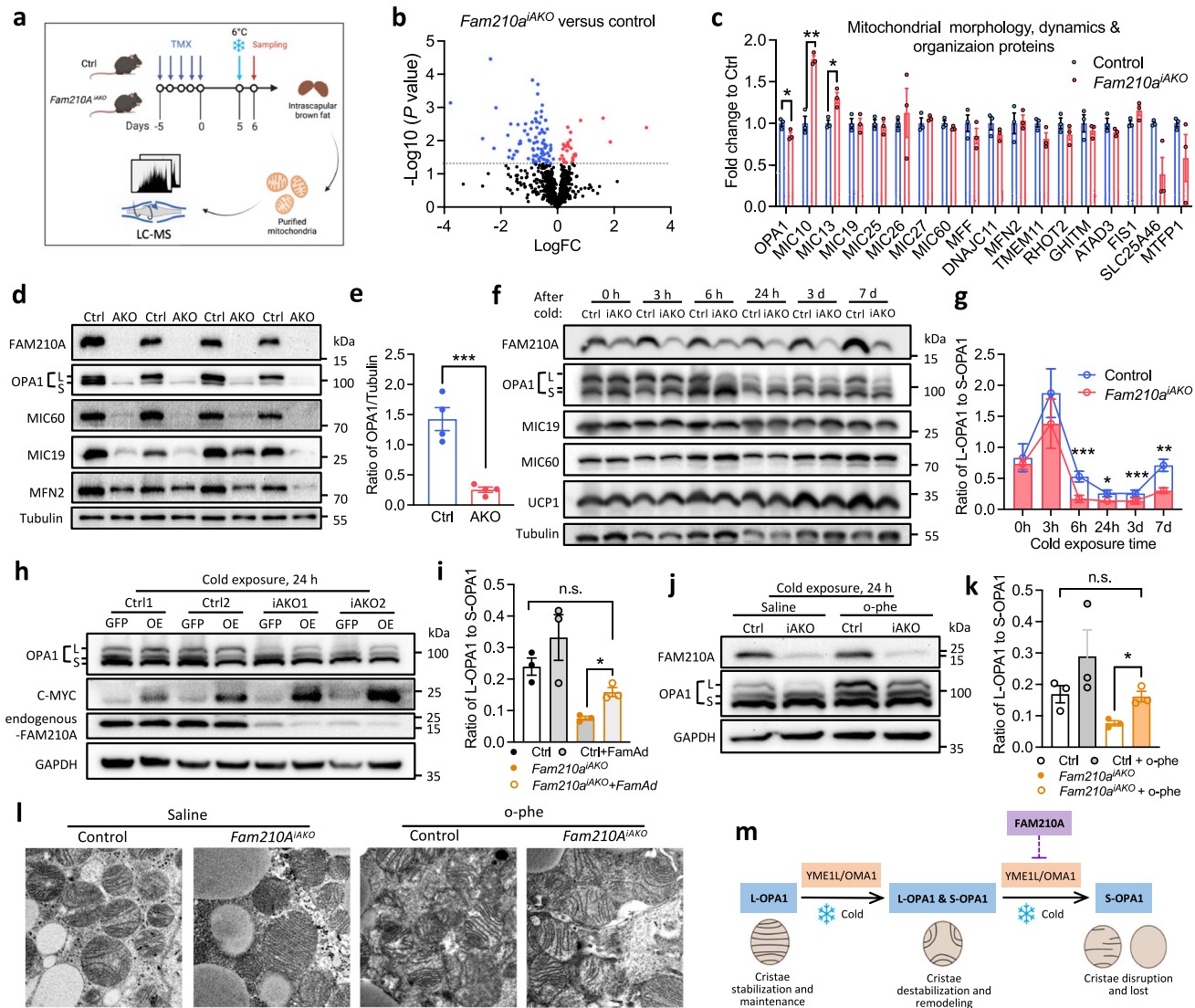

**Fig. 6 | FAM210A regulates OPA1 cleavage through modulating YME1L and OMA1 protease activity. a** Experimental workflow for the study of BAT mitochondrial proteomics of control and *Fam210a^iAKO* mice (diagram created with BioRender.com). **b** Protein abundance differences in BAT mitochondrial proteome between control and *Fam210a^iAKO* mice upon cold exposure for 3 d (FC, fold change; *n* = 3 mice per group; two-tailed unpaired Student's *t* test; differential expression was determined using a cutoff significance level of *P* < 0.05). **c** Expression levels of proteins related to "mitochondrial morphology, dynamics & organization proteins" in BAT mitochondrial proteome of control and *Fam210a^iAKO* mice on cold for 3 d (*n* = 3 mice per group; mean ± s.e.m; two-tailed unpaired Student's *t* test; *P* = 0.049728, 0.001238, and 0.015885). **d** Immunoblotting analysis of mitochondrial structure-related proteins in BAT of control (Ctrl) and *Fam210a^AKO* (AKO) mice upon cold exposure for 7 d (*n* = 4 mice per group). **e** Quantification of total OPA1/tubulin in (**d**) (mean ± s.e.m; two-tailed unpaired Student's *t* test; *P* = 0.001). **f** Immunoblotting analysis of mitochondrial structure-related proteins in control (Ctrl) and *Fam210a^iAKO* (iAKO) mice upon cold exposure for different times (*n* = 5

mice per group for 0 and 24 h; *n* = 3 mice per group for 3 h; *n* = 6 mice per group for 6 h, 3 d, and 7 d). **g** Quantification of L-OPA1/S-OPA1 ratio in (**f**) (mean ± s.e.m; two-tailed paired Student's *t* test; *P* = 0.000306, 0.013745, 0.000536, and 0.002041). **h** Immunoblotting analysis of BAT OPA1 cleavage in Ctrl and iAKO mice with BAT-local delivery of *Fam210a-c-Myc* adenovirus (FamAd) followed by cold exposure for 24 h (*n* = 3 mice per group). **i** Quantification of OPA1 cleavage in (**h**) (mean ± s.e.m; two-tailed paired Student's *t* test; *P* = 0.01615). **j** Immunoblotting analysis of BAT OPA1 cleavage in Ctrl and iAKO mice with BAT-local delivery of o-phenanthroline (o-phe) followed by cold exposure for 24 h (*n* = 3 mice per group). **k** Quantification of OPA1 cleavage in (**j**) (mean ± s.e.m; two-tailed paired Student's *t* test; *P* = 0.01612). **l** Representative TEM images of BAT in mice with treatments in (**j**) (*n* = 3 mice per group; scale bars, 1 μm). **m** Diagram depicting that FAM210A is an essential regulator of YME1L and/or OMA1, and protects L-OPA1 from excessive cleavage during cold exposure (diagram created with BioRender.com). **P* < 0.05, ***P* < 0.01, ****P* < 0.001. Source data are provided as a Source Data file.

## Discussion

Mitochondrial cristae are dynamic bioenergetic compartments whose shape changes in response to external insults and metabolic cues such as cold exposure[4,43]. Cold challenge induces rearrangement of mitochondrial cristae in BAT to facilitate a higher demand of oxidative respiration to maintain body temperature[26,44,45]. However, the molecular mechanism underlying mitochondrial cristae remodeling in BAT

remains unclear. Our present study reports that cold-induced mitochondrial cristae remodeling is intimately linked to mitochondrial proteomic changes. We further show that FAM210A, a poorly characterized protein, is required for cold-induced mitochondrial cristae remodeling and BAT thermogenesis. Mechanistically, FAM210A functions as a molecular brake to prevent excessive cleavage of OPA1 in BAT by modulating YME1L and OMA1 activities. These findings

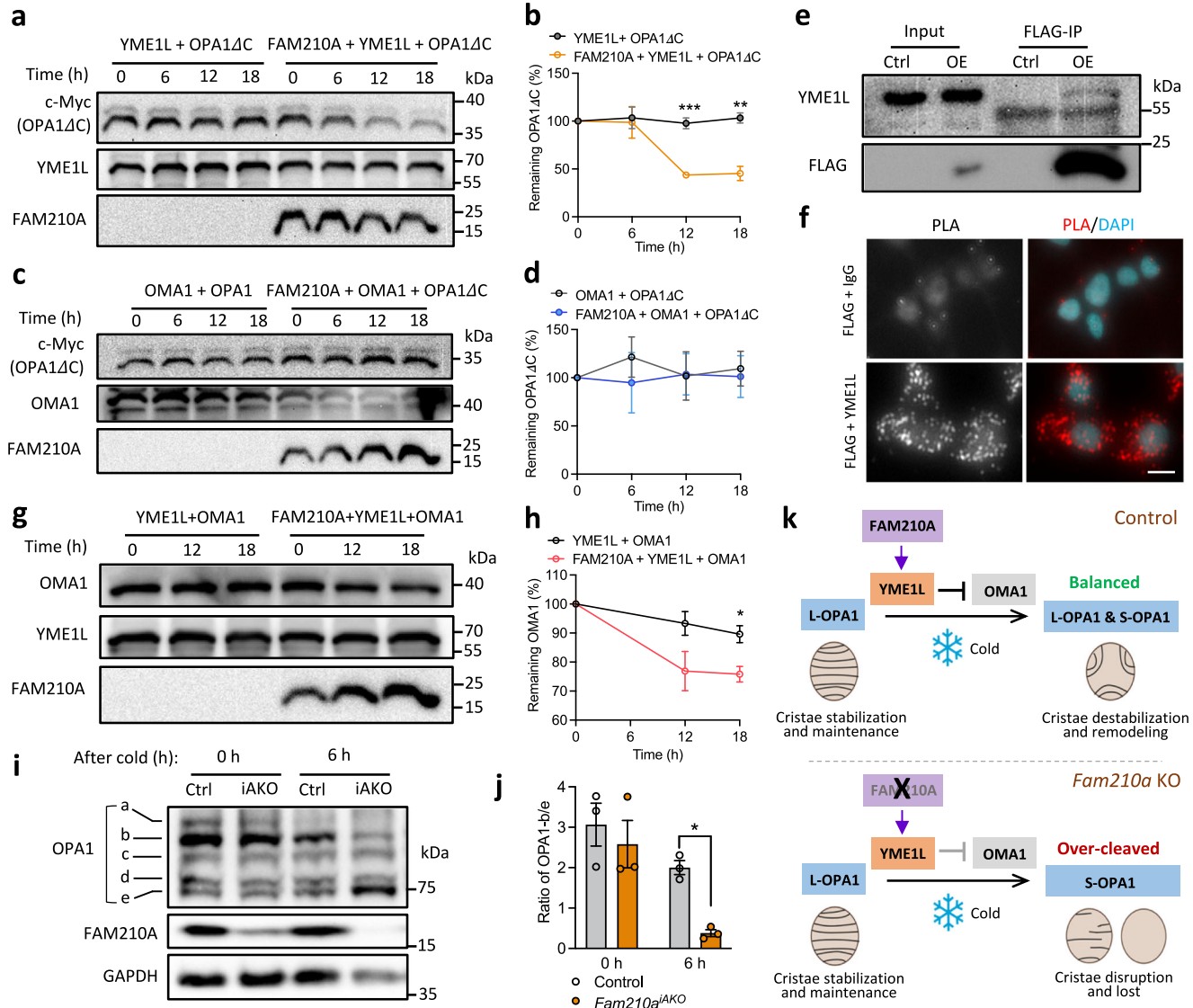

**Fig. 7 | FAM210A interacts with YME1L and modulates its protease activity toward OMA1 and OPA1 cleavage. a** YME1L, OPA1ΔC, and FAM210A were synthesized by *E. coli* cell-free protein synthesis system, and YME1L and OPA1ΔC were incubated with and without the presence of FAM210A (*n* = 3 independent experiments). **b** Quantification of OPA1ΔC protein level in (**a**) (mean ± s.e.m; two-tailed unpaired Student's *t* test; *P* = 0.000762, and 0.003106). **c** OMA1, OPA1ΔC, and FAM210A were synthesized by *E. coli* cell-free protein synthesis system, and OMA1 and OPA1ΔC were incubated with and without the presence of FAM210A (*n* = 3 independent experiments). **d** Quantification of OPA1ΔC protein level in (**c**) (mean ± s.e.m; two-tailed unpaired Student's *t* test). **e** Co-immunoprecipitation (Co-IP) of FAM210A-FLAG and YME1L in *Fam210a-flag* overexpressed HEK293 cell line (*n* = 3 independent experiments). **f** Proximity ligation assay (PLA) showing the protein interaction of FAM210A-FLAG and YME1L in *Fam210a-flag*

overexpressed HEK293 cell line (*n* = 3 independent experiments; scale bar: 10 μm). **g** YME1L, OMA1, and FAM210A were synthesized by *E. coli* cell-free protein synthesis system, and YME1L and OMA1 were incubated with or without the presence of FAM210A (*n* = 3 independent experiments). **h** Quantification of OMA1 protein level in (**g**) (mean ± s.e.m; two-tailed unpaired Student's *t* test; *P* = 0.02497). **i** Immunoblotting analysis of OPA1 in BAT of control and *Fam210a^iAKO* after cold exposure for 0 h and 6 h by using the large format electrophoresis chamber (*n* = 3 mice per group). **j** Quantification of OPA1-b/e in (**i**) (mean ± s.e.m; two-tailed paired Student's *t* test; *P* = 0.02418). **k** Diagram depicting that FAM210A enhances YME1L activity thus suppressing OMA1, which protects L-OPA1 from over-cleavage during cold exposure to facilitate mitochondrial cristae remodeling (diagram created with BioRender.com). **P* < 0.05, ***P* < 0.01, ****P* < 0.001. Source data are provided as a Source Data file.

demonstrate the importance of mitochondrial cristae remodeling in thermogenic adipose tissue and uncover a physiological function of FAM210A in adipose tissue.

We observe the sophisticated remodeling of mitochondrial cristae ultrastructure in BAT during cold exposure, which triggers the rearrangement of unstructured cristae into multiple lamellar cristae within 7 d. Cristae membrane morphology influences the organization and function of proteins localized on mitochondrial inner membrane, such as the OxPhos system, to directly impact cellular metabolism[43,46]. In BAT, increasing cristae density elevates oxidative rates to provide thermogenic energy in response to cold. Prior to the increase of cristae

density, we noticed a reduction of cristae density in the early stages of cold exposure. This destabilization of cristae structure may be necessary for proper cristae remodeling to increase the density and formation of the multi-lamellar structure. The newly formed cristae will then be stabilized. Stabilization of cristae is essential for controlling mitochondrial ROS levels[47,48]. Previous studies have shown that a substantial increase of mitochondrial ROS is required for acutely activating thermogenesis in BAT[15,31,49]. In this regard, mitochondrial cristae reduction in the early stages of cold exposure may function to increase ROS levels to stimulate thermogenesis. This notion is also supported by the observed L-OPA1 cleavage within 24 h upon cold

exposure, which destabilizes cristae structure and promotes ROS generation[29,50]. In addition, mitochondrial cristae play a role in many other processes beyond ROS regulation[5]. The biological function of Class 2 mitochondria at the early stages of cold exposure warrants further investigation.

The mitochondrial proteome is of dual genetic origins. More than 98% of mitochondrial proteins are encoded by nuclear DNA, synthesized by cytosolic ribosomes, and subsequently imported into mitochondria. Additionally, an increasing number of proteins are identified to be multi-localized to several cellular compartments[27,51]. Therefore, mitochondrial proteomics provides a higher specificity over whole-cell proteomics to determine the abundance and dynamics of mitochondrial proteins. Our study represents a pioneering study to examine temporal dynamics of the cold-induced mitochondrial proteome in BAT. Purified mitochondrial proteomics yields a limited number of proteins[52,53], and a high-resolution mitochondrial proteome has not been established for adipose tissue. Therefore, to maximize the coverage, we performed proteomics with crude mitochondrial fractions. As mitochondria are known to intertwine with other organelles, such as ER, lipid droplets, and peroxisomes, the crude mitochondrial proteome may be overrepresented. Nonetheless, ~60% of the proteins identified by our proteomic analysis are bona fide mitochondrial proteins according to MitoCarta3.0 and MitoCoP[27,54].

A recent study reported that FAM210A is highly expressed in skeletal muscle, heart, and brain, indicating its potential role in these mitochondria-rich tissues. Indeed, the global *Fam210a* KO in mice led to embryonic lethality (E9.5 day)[55], suggesting the requirement of FAM210A for organogenesis and tissue development. Moreover, a phenotypic study showed that tamoxifen-inducible *Fam210a* KO in juvenile skeletal muscle resulted in muscle weakness and mildly decreased lean mass[55]. Still, the physiological role of FAM210A and its molecular function in mitochondria remains poorly understood. Our study discovered a previously unappreciated role of FAM210A in the development and metabolism of brown adipose tissue. Our data demonstrate that FAM210A is required for cold-induced mitochondrial cristae remodeling, adaptive thermogenic activity, and BAT characteristics. A recent study reported that FAM210A regulates mitochondrial protein translation in cardiac muscle by interacting with the translation elongation factor EF-TU[33]. However, we did not observe any differences in the levels of BAT mitochondrial-encoded proteins in TMX-induced *Fam210a* KO mice, even after 7 d of cold exposure. These data suggest that FAM210A plays cellular context-dependent roles in different cells. While it mediates mitochondrial protein translation in cardiac myocytes that requires substantial ATP production to power heart beating, it mediates OPA1 cleavage to facilitate cristae remodeling in BAs in response to cold-induced thermogenesis that uncouples ATP synthesis from electron transport.

The dynamic balance between the membrane-anchored L-OPA1 (which mediates cristae formation to facilitate fusion) and the soluble S-OPA1 (which induces cristae destabilization to facilitate fission) is crucial for mitochondrial remodeling under various physiological conditions[29,30,56]. Our observations reveal that the ratio of L-OPA1 and S-OPA1 is dedicatedly regulated by FAM210A in BAT mitochondria during cold exposure to govern the dynamic mitochondrial cristae remodeling. To facilitate mitochondrial remodeling and augment thermogenic function, acute cold exposure induces rapid mitochondrial fragmentation in BAT[25]. Given that OMA1 deletion decreases the thermogenic activity of BAT, OMA1-mediated cleavage of L-OPA1 to S-OPA1 is required for cold-induced mitochondrial remodeling[24]. On the other hand, abnormal accumulation of S-OPA1, due to excessive activation of YME1L and/or OMA1, causes mitochondrial damage and cell death[29]. Thus, either too little or too much cleavage of OPA1 is detrimental, and the cleavage of L-OPA1 must be tightly controlled to maintain the dynamic balance of L-OPA1 and S-OPA1 in response to cold

exposure. Our study sheds light on this process, demonstrating that FAM210A directly enhances YME1L activity to finely tune OMA1-mediated OPA1 cleavage.

We propose that the cold-inducible FAM210A functions as a molecular brake to modulate YME1L activity on OMA1, preventing excessive cleavage of OPA1 by OMA in BAT. In support of this, the loss of *Fam210a* led to abnormal OMA1-mediated OPA1 processing at 6 h, indicating that induction of FAM210A is necessary for maintaining OMA1 activity at a proper level during the early stages of cold-induced mitochondrial remodeling. Moreover, after the maximal cleavage of OPA1 at 24 h, the robust increase of FAM210A level serves to prevent further cleavage of OPA1, enabling the restoration of L-OPA1 at 3 d and 7 d as cristae remodeling is completed. While the kinetics of L-OPA1 and S-OPA1 and its regulatory proteins remain to be determined, we speculate that the proper suppression of OMA1 activity by FAM210A-YME1L axis is a protective strategy for stepwise OPA1 cleavage to warrant the kinetics of OPA1. Collectively, we have defined a previously unknown regulatory pathway (FAM210A-YME1L/OMA1) that acts as a molecular brake to finely tune the OPA1 cleavage during cold-induced mitochondrial remodeling in BAT.

In addition to mediating cristae remodeling, emerging studies have demonstrated that OMA1 acts as a mitochondrial stress sensor that relays the signal to the cytosol and activates integrated stress response (ISR) by cleaving DELE1[57–59]. In this regard, overactivation of OMA1 in *Fam210a* KO BAT may stimulate mitochondrial ISR, thus reducing cytosolic protein synthesis globally and attenuating mitochondrial biogenesis[57,60,61]. Considering the lag between deficient cristae remodeling (at 6 h) and impaired BAT thermogenesis and FAO in inducible *Fam210a* KO mice (at 7 d), it is possible that the accumulation of OMA1-induced ISR gradually compromises mitochondrial biogenesis in the inducible KO mice. Thus, in addition to regulating mitochondrial cristae remodeling, FAM210A may also function to restrain OMA1 activity to prevent mitochondrial ISR and allow mitochondrial biogenesis during long-term cold exposure.

We observed that the reaction of FAM210A-mediated YME1L activity towards OMA1 degradation is much slower in the cell-free system than that observed in vivo. This may be due to the lack of many other components normally present in vivo in our cell-free protein interaction system[62]. Specifically, the cell-free system employed in the present study did not involve a reconstituted mitochondrial membrane, which may limit YME1L, OMA1 and FAM210A assembly and activity due to the insoluble transmembrane domain[63]. In support of this speculation, several studies have reported that the ATPase and protease activity of YME1L relies on the hexamerization of its catalytic domain on the mitochondrial inner membrane[63–65]. Future research is expected to dissect the in-depth molecular mechanisms underlying FAM210A function in the regulation of YME1L/OMA1 activity and cristae remodeling by using a proteoliposome system[66,67].

## Methods

### Ethical statement

This research complies with all relevant ethical regulations. All procedures involving animals were done in compliance with National Institutes of Health and Institutional guidelines with approval by the Purdue Animal Care and Use Committee.

### Animals

The mouse lines used in this study were in a C57BL/6J background. *Fam210a*^*flox/flox*^ mice were generated using the CRISPR-Cas9 technique. Two guide RNAs (CCAGAATGCCCTGTGATGAA, CACAAAGGGACA TGACAGGT) were microinjected with Cas9 enzyme and donor vector into fertilized eggs to insert LoxP sites flanking exon 2 and exon 4 of *Fam210a* in the C57BL/6J strain at Nanjing Biomedical Research nstitute of Nanjing University. *Adipoq*^*Cre*^ (JAX stock #010803) and *Adipoq*^*CreER*^ (JAX stock #024671) mouse stocks were purchased from

the Jackson Laboratory. Mouse genotyping was performed by genomic DNA isolation from ear and was screened by polymerase chain reaction (PCR) using primers and the program following the protocols provided by the supplier. The genotypes of experimental knockout and corresponding control mice are as follows: *Fam210a^AKO* (*Adipoq^Cre*; *Fam210a^flox/flox*), *Fam210a^iAKO* (*Adipoq^CreER*; *Fam210a^flox/flox*), and control (*Fam210a^flox/flox*). Mice were housed and maintained in the animal facility with free access to water and standard rodent chow food (2018 Teklad Global 18% Protein Rodent Diets), and were housed under 12-h light/dark cycle, at 22 °C, and 45% humidity on average. For all animal-based experiments, at least three pairs of gender-matched littermates at the age of 2–5 months were used. Mice were anesthetized with 3% isoflurane followed by euthanasia by cervical dislocation.

### Cold and thermoneutral exposure experiments
Cold exposure experiments were performed on 2- to 4- month-old adult mice in a temperature-controlled chamber (Memmert, HPP260). For chronic cold exposure, mice were individually caged with minimal bedding and free access to food and water at 6 °C for one week. For the acute cold challenge, mice were transferred to 6 °C on fasting, and body temperature was monitored every 30 min for the first hour and then measured every hour for 5 h. Cold challenge experiments started at or around ZT6 (12:00 pm). The intrarectal temperature of mice was monitored using a digital thermometer (ETI, MicroTherma2) in combination with a copper thermocouple probe (TypeT). The probe was inserted 2 cm or 1.6 cm into the anal ducts of male and female adult mice, respectively. For tissue temperature recording, mice under anesthesia were implanted with type T thermocouple microprobes in iBAT, and the temperature was recorded by Iso-Thermex (Columbus Instruments). For thermoneutral acclimation, mice were housed at 30 °C for two weeks. A FLIR infrared camera (FLIR Systems, FLIR T440) was employed to capture thermography.

### Local delivery of adenovirus and o-phenanthroline in interscapular brown adipose tissue
iBAT local delivery of adenovirus and o-phenanthroline (o-phe) (J.T.Baker, cat#T170-02) were performed following the published protocol[68]. In short, adenovirus was injected into the iBAT of control and *Fam210a^iAKO* mice at a viral titer of $3 \times 10^{11}$ particles per mouse. Sixty microliters of adenovirus at a dose of $2.5 \times 10^{12}$ particles per ml were injected into each iBAT lobe. O-phe was prepared in saline at a concentration of 100 nM and was injected into the iBAT of control and *Fam210a^iAKO* mice at a dose of 10 nmol per mouse (5 nmol per BAT lobe). The mice were subjected to further experiments 7 d after injection.

### In vitro cell culture
Isolated primary BAT stromal vascular fraction (SVF) cells and brown adipocyte cell line (a gift from Dr. Shingo Kajimura) were cultured in Dulbecco's Modified Eagle's Medium F12 (DMEM F12, Gibco) supplemented with 10% FBS (HyClone), 1% penicillin/streptomycin (P/S, HyClone) at 37 °C with 5% $CO_2$ for 2 d, followed by replacing with fresh medium every 2 d. When reached 100% confluence, primary BAT SVF cells were induced to adipogenic differentiation with induction medium (DMEM supplemented with 10% FBS, 1% P/S, 2.85 μM insulin, 0.3 μM dexamethasone, 1 μM rosiglitazone, and 0.63 mM 3-isobutyl-methylxanthine) for 4 d, and then differentiated in differentiation medium (DMEM, 10% FBS, 1% P/S, 200 nM insulin and 10 nM T3) for 3 to 4 d until adipocyte mature. Lenti-X 293 T (TaKaRa, 632180), 293 A cell line (Thermo Fisher, R70507), HEK293 cell line (ATCC, CRL-1573), and Cos-7 (ATCC, CRL-1651) cell lines were cultured in DMEM supplemented with 10% FBS and 1% P/S. Mycoplasma was certified when cells were purchased, and cell identity was authenticated by morphological features.

### Tamoxifen (TMX) administration
TMX (Biosynth Carbosynth, cat#FT27994) was prepared in corn oil at a concentration of 10 mg ml⁻¹, and experimental and control mice were injected intraperitoneally with 2 mg TMX per day per 20 g body weight for 5 d to induce Cre-mediated deletion. TMX injections were initiated on adult mice, and experimental mice were used at the time stated in the text.

### 2-NBDG administration and imaging
2-NBDG (3 mM in PBS, 0.3 mL; APEXBIO, cat#B6035) or PBS (negative control) was administered to mice at 10–12 weeks of age via a tail-vein injection after 2 h cold treatment with fasting, and mice were exposed in cold for another 1 h before imaging[69,70]. Images were taken with the Ami optical imaging system (Spectral Instruments Imaging). After 3 h cold treatment, mice were immediately euthanized and dissected. Multiple tissues, including the BAT, iWAT, tibialis anterior (TA) muscle, quadriceps (QU) muscle, gastrocnemius (GAS) muscle, liver, heart, spleen, and kidney, were collected, followed by the ex vivo imaging (ex/em: 465/530 nm). The fluorescent images were analyzed by AURA Imaging Software (Spectral Instruments Imaging).

### Body composition analysis
Body weight and body composition were measured in *ad-lib* fed conscious mice using EchoMRI 3-in-1 system nuclear magnetic resonance spectrometer (Echo Medical Systems) to determine whole-body lean and fat mass.

### Indirect calorimetry study
Oxygen consumption ($VO_2$), carbon dioxide production ($VCO_2$), and energy expenditure were measured using an indirect calorimetry system (Columbus Instruments, Oxymax) installed under a constant environmental temperature (22 °C) and a 12-hour light/dark cycle. Mice in each chamber had free access to food and water. CL316,243 was intraperitoneally injected into mice after 24 h of measurements and at a dose of 1 mg/kg body weight (in saline). Obtained indirect calorimetry data were analyzed by CaIR-ANCOVA (https://calrapp.org/), a regression-based analysis of energy expenditure in mice[71].

### Blood glucose measurements
For glucose-tolerance test (GTT), mice were given an intraperitoneal (i.p.) injection of 100 mg ml⁻¹ D-glucose (2 g/kg body weight) after overnight fasting, and tail blood glucose concentrations were measured by a glucometer (Accu-Check Active, Roche). For insulin-tolerance test (ITT), mice were fasted for 4 h before i.p. injection of insulin (Sigma Aldrich, cat#I6634) (0.75 U/kg body weight), and tail blood glucose concentrations were monitored. For both GTT and ITT, each mouse was singly caged with blinded cage numbers and random orders.

### Measurement of serological parameters
Serum levels of non-esterified fatty acids (NEFA), cholesterol, high-density lipoprotein (HDL), low-density lipoprotein (LDL), and triglycerides were examined by standard kits from Randox Laboratories.

### Ex vivo BAT treatment
Upon dissection, BAT was washed with DMEM 3 times and cut into ~1 mm³ blocks. Five tissue blocks of BAT were added to one well of a 12-well plate and incubated with DMEM containing 20% FBS and 1% P/S. For o-phe treatment, ex vivo BAT was treated with 2 mM o-phe for 0 min to 60 min. The treatments were immediately performed after BAT dissection.

### Brown adipocyte (BA) isolation
Primary BAT SVF cells were isolated using collagenase digestion, followed by density separation. Briefly, the iBAT was minced and digested

in 1.5 mg/ml type I collagenase (Worthington, cat#LS004197) at 37 °C for 40 min. The digestions were terminated with DMEM containing 10% FBS and filtered through 100-μm cell strainers to remove connective tissues and undigested trunks of tissues. Cells were then centrifuged at $450 \times g$ for 5 min to separate the SVF cells in the sediment. The freshly isolated SVF cells were seeded and cultured in growth medium.

Primary mature BAs were collected as previously described[72]. In short, iBAT was dissected, minced, and digested in 2 mg/ml type I collagenase at 37 °C for 30 min. The digestions were terminated with DMEM containing 10% FBS. The mixture was centrifuged at $30 \times g$ for 10 min, and the infranatant and SVF were removed with a syringe. BAs were washed with DMEM containing 10% FBS 3 times and directly subjected to fixation and imaging studies.

## Mitochondrial isolation
Mitochondria were isolated from the whole iBAT of mice at 10 weeks of age as previously reported[73]. Briefly, iBAT were washed three times in PBS, cut into pieces, and homogenized with a Glass/Teflon Potter Elvehjem homogenizer (5 mL) grinder in 2 ml $IBm_1$ buffer (67 mM sucrose, 50 mM Tris/HCl, 50 mM KCl, 10 mM EDTA, and 0.2% BSA in distilled water, pH 7.4) for at least 20 strokes. The homogenate was centrifuged at $8500 \times g$ for 10 min at 4 °C, and the fat was discarded with the supernatant. The pellet was suspended with 1 ml $IBm_1$ buffer and centrifuged at $700 \times g$ for 10 min at 4 °C. The supernatant containing mitochondria was centrifuged again at $8500 \times g$ for 10 min at 4 °C, and the sediment was collected. The sediment mitochondria were resuspended in 100 μl $IBm_1$ buffer, and the concentration was determined using Pierce BCA Protein Assay Reagent (Pierce Biotechnology, cat#23225) and NanoDrop 2000c (Thermo Fisher Scientific).

## Oxygen consumption rate (OCR) assays
OCR was measured using Seahorse XFe24 Analyzer (Agilent Technologies). iBAT-derived SVFs were seeded in matrigel-coated XF24 plates at 50 K per well and differentiated for 8 d. On the day of experiments, the cells were treated with 10 μM CL316,243 for 6 h, then washed three times and maintained in XF base medium (Agilent Technologies, cat#102353-100) supplemented with 1 mM sodium pyruvate, 1 mM glutamine, and 5 mM glucose for 1 h in a 37 °C non-$CO_2$ incubator before the assay. Basal OCR was recorded 3 times (3 min recording and 3 min mixing cycles), followed by sequential injections of oligomycin (3 μM), FCCP (3 μM), and rotenone/antimycin A (1 μM). OCR was recorded in 3 cycles of 3 min mixing followed by 3 min recording after each injection. The basal OCR was normalized to cell number.

## Glycolytic rate assay
Glycolytic rate was measured using Seahorse XFe24 Analyzer. The cells were plated and treated as described for the OCR assay. The medium was changed to XF base medium supplemented with 2 mM glutamine. The extracellular acidification rate (ECAR) was measured in 3 cycles of 3 min mixing and 3 min recording sequential injection of glucose (25 mM), oligomycin (3 μM), and 2-DG (100 mM). Recording was performed three times after each injection, following the same mixing and measuring cycles. Glycolytic rate was calculated using the Seahorse Glycolysis Rate assay report generator (Agilent Technologies).

## Adenovirus-mediated overexpression
The adenovirus was generated using the AdEasy system[74]. The coding sequence of *Fam210a-Myc* was cloned from pCMV6-Fam210a-Myc-DDK plasmid, purchased from origene (Origene, cat#MR203660). The cloned sequence was inserted into the pAdTrack-CMV plasmid (Addgene, plasmid#16405). The formed pAdTrack-Fam210a (pAd-Track-CMV as the control) plasmid was digested by PmeI (NEB), and then transfected into BJ5183-competent cell with pAdEasy-1 (Stratagene, cat#BJ5183-AD-1)[74]. The positive recombinant plasmid was

detected by PacI (NEB) digestion. For adenovirus generation, 293A cells (60–70% confluent) in 10-cm culture dishes were transfected with 4 mg of PacI-digested recombinant plasmid using Lipofectamine 2000 (Invitrogen, cat#11668030) according to the manufacturer's protocol. After 10 d of transfection, the cells were collected, and recombinant adenovirus was released by the freeze-thaw method. To increase the titer of adenovirus, two more rounds of infection were adapted to amplify the recombinant virus, and then the titer was determined by the expression of GFP. To apply the adenovirus in animal experiments, the virus was purified by CsCl (Fisher Scientific, cat#BP1595-1) gradient centrifugation and dialysis.

## Vector construction and cell transfection
The *Fam210a-flag* overexpressed and empty vector control HEK293 cells were generated using pLJM-EGFP (Addgene #19319). The coding sequence of *Fam210a-Myc* was cloned from pCMV6-Fam210a-Myc-DDK plasmid. *Fam210a-flag*-pLJM (or pLJM-EGFP), psPAX2 (Addgene #12260), and pMD2.G (Addgene #12259) plasmids were co-transfected with a ratio of 5:4.5:0.5 into Lenti-X 293T cells using Lipofectamine 2000. The virus was harvested 36 h and 48 h post-transfection and filtered through a 0.45-μm pore size filter. To generate *Fam210a-flag* expressed stable HEK293 cells, the cells were infected with lentivirus of *Fam210a-flag* with 10 μg/mL polybrene. Puromycin (1 μg/mL) was used to select the infected cells 1 d post-infection.

## Single-molecule localization microscopy
Single-molecule localization microscopy (SMLM) imaging was performed on a custom-built setup on an Olympus IX-73 microscope stand (Olympus America, IX-73) equipped with a 100×/1.35-NA silicone-oil-immersion objective lens (Olympus America, UPLSAPO100XS) and a PIFOC objective positioner (Physik Instrumente, ND72Z2LAQ). Samples were excited by a 642 nm laser (MPB Communications, 2RU-VFL-P-2000-642-B1R), which passed through an acousto-optic tunable filter (AA Opto-electronic, AOTFnC-400.650-TN) for power modulation. The excitation light was focused on the pupil plane of the objective lens to provide widefield or highly inclined and laminated optical sheet illumination. For conventional single-molecule imaging, the fluorescence signal passed through a quadband dichroic mirror (Chroma, ZT405/488/561/647rpc) and then through a bandpass filter (Semrock, FF01-731/137-25) before detection. It was magnified by relay lenses arranged in a 4 f alignment to a final magnification of ~54 at a scientific complementary metal-oxide-semiconductor (sCMOS) camera (Hamamatsu, Orca-Flash4.0v3) with an effective pixel size of 119 nm. For ratiometric multicolor imaging, the salvaged fluorescence signal was reflected by the dichroic mirror (Chroma, ZT405/488/561/647rpc), and then was reflected by another dichroic mirror (Semrock, FF654-SDi01-25×36). After being filtering by a bandpass filter (Semrock, FF01-661/20-25), the salvaged fluorescence signal was then detected by an sCMOS camera (Hamamatsu, Orca-Flash4.0LT) operated in sync with the camera detecting conventional fluorescence. The intensity ratio between two fluorescence channels was used to identify the dye of each emitter. The imaging system was controlled by custom-written LabVIEW (National Instruments) programs.

The coverslip with cells on top was placed on a custom-made holder. 100 μL of single molecule super-resolution imaging buffer (10% (w/v) glucose in 50 mM Tris, 50 mM NaCl, 10 mM β-Mercaptoethylamine hydrochloride (M6500, Sigma-Aldrich), 50 mM 2-Mercaptoethanol (M3148, Sigma-Aldrich), 2 mM cyclooctatetraene (138924, Sigma-Aldrich), 2.5 mM protocatechuic acid (37580, Sigma-Aldrich) and 50 nM protocatechuate 3,4-Dioxygenase (P8279, Sigma-Aldrich), pH 8.0) were added on top of the coverslip. Then another coverslip was placed on top of the imaging buffer. This coverslip sandwich was then sealed with two-component silicon dental glue (Dental-Produktions und Vertriebs GmbH, picodent twinsil speed 22). The sample was first excited with the 642-nm laser at a low intensity of

~50 W/cm² to find a region of interest. Before fluorescence imaging, bright-field images of this region were recorded over an axial range from −1 to +1 μm with a step size of 100 nm as reference images for focus stabilization. Single-molecule blinking data were then collected at a laser intensity of 2–6 kW/cm² and a frame rate of 50 Hz. Imaging was conducted for ~30 cycles with 2000 frames per cycle. Calculated using Cramer-Rao lower bound (CRLB), the localization precisions were 11.67 ± 3.68 nm (mean ± std, $N = 133{,}299$) for Fam210a-c-Myc-CF660C and 12.17 ± 4.05 nm (mean ± std, $N = 85{,}189$) for Tom20-Alexa Fluor 647.

## Transmission electron microscopy (TEM)

iBAT and iWAT were dissected, cut into 1 mm³ blocks, and fixed immediately in the fixative buffer (2.5% glutaraldehyde, 1.5% paraformaldehyde (PFA) in 0.1 M cacodylate buffer). Samples were rinsed in deionized water followed by fixation in 2% osmium tetroxide for 1 h. Then, the samples were washed in deionized water, followed by fixation in 1% uranyl acetate for 15 min. The samples were then dehydrated with a series of graded ethanol followed by dehydration in acetonitrile and embedded in epoxy resin (EMbed 812: DDSA: NMA 5:4:2; 0.22 DMP-30). Ultrathin sections were cut at 70 nm and stained with uranyl acetate and lead citrate. Stained sections were examined under Tecnai T12 transmission electron microscope attached with a Gatan imaging system. The images were processed using the IMOD tomography package[75].

## Immunofluorescence staining

Immunofluorescence was performed in cultured BAs and the Cos-7 cell line. Briefly, samples were fixed in 4% PFA for 10 min, quenched with 100 nM glycine for 10 min, and permeabilized and blocked in blocking buffer (PBS containing 5% goat serum, 2% bovine serum albumin (BSA), 0.2% Triton X-100, and 0.1% sodium azide) for 1 h at RT. Samples were subsequently incubated with primary antibodies diluted in blocking buffer overnight at 4 °C. After washing with PBS, samples were incubated with secondary antibodies and DAPI (Invitrogen, cat#D1306) for 1 h at RT. Mature BAs were stained with Bodipy 493/503 (Invitrogen, cat#D3922) and DAPI. For MitoTracker (Cell Signaling Technology, cat#9082) staining, live cells were incubated with FBS-free DMEM containing 0.2% MitoTracker for 50 min, then fixed with iced-cold methanol for 20 min. Antibodies used for immunofluorescence staining are listed in Supplementary Table 1. Primary antibodies and dilutions were used as follows: C18orf19 (FAM210A, Invitrogen, PA5-53146; 1:300), c-Myc (Santa Cruz Biotechnology, sc-40; 1:300), Tom20 (Santa Cruz Biotechnology, sc-11415; 1:300). Fluorescent images were captured using a Leica DM6000B microscope.

## H&E staining and immunohistochemistry staining

Adipose tissues from control, *Fam210a^AKO* and *Fam210a^iAKO* mice were dissected and fixed in 4% PFA for 24 h at RT. Then the tissues were dehydrated with a series of graded ethanol, cleared with xylene, and embedded into paraffin. The samples were cut into 4-μm thick slices, deparaffinized, and rehydrated using xylene, ethanol, and water by standard methods. For H&E staining, sections were stained with haematoxylin for 15 min, then rinsed in running tap water and stained with eosin for 1 min and 30 min for BAT and WAT samples, respectively. For immunohistochemistry staining, antigen retrieval was performed by submerging slides in 0.01 M sodium citrate (pH 6.0) and heated to 96 °C for 20 min in a laboratory microwave (PELCO). Slides were incubated with 3% hydrogen peroxide and 2.5% normal horse serum (S-2012, Vector), followed by incubation with rabbit polyclonal anti-UCP1 primary antibody (Abcam, cat#ab10983) diluted 1:200 in 2.5% normal horse serum (Vector, cat# S-2012) for 60 min. Signals were detected with a rabbit IgG horseradish peroxidase (HRP)-conjugated secondary antibody (Jackson ImmunoResearch, cat#111-035-003). Labeling was visualized with 3,3′-diaminobenzidine (DAB)

(Acros Organics, cat#112090050) as the chromogen (Vector, cat#SK-4105). The nuclei were counterstained with hematoxylin. Primary antibodies and dilutions were used as follows: UCP1 (Abcam, ab10983; 1:200). All the images were captured using a Leica DM6000B microscope. The images shown are representative results of at least three biological replicates.

## Proximity ligation assay (PLA)

After incubation with primary antibodies as described in immunofluorescence staining, PLA was performed in a 20 μl reaction system per step per slide according to the Sigma-Aldrich Duolink In Situ PLA manufacturer's instructions (catalog # DUO92101). In brief, two PLA secondary probes, anti-mouse MINUS and anti-rabbit PLUS, were diluted 1:5 in antibody diluent buffer provided by the manufacturer and incubated overnight at 4 °C. Slides were washed once in wash buffer A for 10 min. For the following ligation reaction, PLA ligation stock was diluted 1:5 in distilled H₂O with ligase (added at 1:40), and then slides were incubated for 1 h at 37 °C. Slides were then washed in wash buffer A for 5 min and the PLA amplification reaction buffer was added (1:5 amplification stock and 1:80 polymerase diluted in dH₂O). Slides were incubated for 2 h at 37 °C, then washed with 1x wash buffer B for 10 min, 0.01x wash buffer B for 1 min, and 1× PBS for 1 min. After staining with DAPI, slides were ready for image using Leica DM6000B microscope. Primary antibodies and dilutions were used as follows: normal rabbit IgG (Cell signaling, 2729; 1:300), FLAG (Sigma, F1804; 1:500), YME1L (Proteintech, 11510-1-AP; 1:300).

## Flow cytometry analysis of isolated mitochondria

Isolated mitochondria were stained with MitoTracker green (Invitrogen, cat#M7514) and tetramethylrhodamine ethyl ester (TMRE; Cayman chemical, cat#21426) for 10 min on ice before analysis on a BD LSRFortessa with BD FACSDiva software. MitoTracker green was detected in the FITC channel and TMRE in the PE channel. FlowJo V 10.8 software (FlowJo) was used for analysis.

## Protease protection assay

Isolated mitochondria from iBAT were subjected to proteinase K (Omega Bio-Tek, cat#AC116) (100 μg/ml) to digest exposed proteins. Osmotic shock (25 mM sucrose, 10 mM MOPS-KOH, pH 7.2) was used to disrupt the mitochondrial outer membrane. TritonX-100 was added at a final concentration of 0.5% (w/v) to disrupt the mitochondrial inner membrane. Digestion reactions were performed for 20 min on ice, and proteinase K activity was then blocked with PMSF (2 mM). Mitochondrial fractions were subjected to SDS-PAGE and detected by immunoblotting.

## DNA extraction and relative mitochondria DNA copy number

iBAT was digested in 500 μl DNA extraction buffer (10 mM Tris-HCl (pH 8.0), 1% SDS, 50 mM EDTA, 50 μg proteinase K) overnight at 55 °C. After digestion, genomic DNA was extracted using a phenol-chloroform method. In brief, an equal volume of phenol was added to the sample, followed by centrifugation at $12{,}000 \times g$ for 20 min at 4 °C to generate different liquid phases. The top phase containing DNA was transferred to a new tube, and the same volume of phenol: chloroform: isoamyl solution (25: 24: 1 ratio) was added, followed by a 15-min spin with the same settings. The same volume of a chloroform: isoamyl solution (24:1 ratio) was added to the top layer and spun at $10{,}000 \times g$ for 15 min. DNA was precipitated from the top layer by adding 1/10 volume of 3 M sodium acetate, 2X volume of precooled 100% ethanol and pelleted by centrifugation at $10{,}000 \times g$ for 10 min. The supernatant was aspirated, and 30 μl dH₂O was added to each remaining pellet. The concentration and quality of genomic DNA were measured by Nanodrop 2000c and subjected to further real-time PCR analysis. To quantify the number of mitochondria DNA (mtDNA), we used the following *Cytochrome b* (*CYTB*) primers: Forward:

TCATCGACCTCCCCACCCCATC; Reverse: CGTCTCGAGTGATGTG GGCGATT. To quantify nuclear DNA, we used a primer set that detects *β−2-microglobulin* (*B2M*): Forward: TGGCCATACTACCCTGAATGAG TCC; Reverse: ATGTATTGTGCAATGCTGCTGCTCG.

## Fatty acid oxidation (FAO) assay

FAO activity was measured in iBAT lysate by using [1-$^{14}$C] palmitate (Moravek, cat#MC121) and a liquid scintillation counter[76]. Briefly, upon dissection, iBAT tissue was weighed, washed in cold PBS three times, and minced. 9X the volume of each iBAT sample was added in SET buffer (250 mM Sucrose, 1 mM EDTA, 10 mM Tris-HCl, pH 7.4), followed by homogenization. Homogenized lysate (450 μl) was placed into a new tube, and the total volume was brought to 2 ml with SET buffer. 80 μL of each diluted sample was added to 120 μl Oxidation buffer (0.5% BSA, 0.000625 μCi [1-$^{14}$C] palmitate, 200 μM palmitate, 2X Reaction Buffer (125 mM Sucrose, 25 mM $K_2HPO_4$, 20 mM KCl, 2.5 mM $MgCl_2$−6$H_2O$, 2.5 mM L-Carnitine, 0.25 mM Malic Acid, 20 mM Tris-HCl, 2.5 mM DTT, 4 mM ATP, 0.125 mM CoA, pH 7.4)). Each tube was capped with a $CO_2$ trap-rubber stopper to prevent air leaks. Tubes were incubated at 37 °C for 30 min with gentle shaking. 200 μl HCl was added to the bottom of the tube, avoiding the center well, to stop the reaction. 300 μl phenethylamine (Acros Organics, cat#156491000) was added to the center well. $CO_2$ was then allowed to be trapped by incubating the tubes at RT for 1 h with shaking. The $CO_2$ trap caps were removed, and the center well was cut into a scintillation vial. Ecolite (MP Biomedicals, cat#882475) was added, and the scintillation was counted by TRI-CARB 1600 liquid scintillation Counter (Packard).

## Mitochondrial ATP detection

Mitochondrial ATP level was measured using an ATP Assay Kit (Cayman Chemical, cat#700410). Mitochondria isolated from iBAT were rinsed with cold PBS and lysed with ice-cold 1X ATP Detection Sample Buffer. Mitochondria (20 μg) and standards were plated into a 96-well plate. 100 μL of freshly prepared reaction mixture (1X ATP Detection Sample Buffer, D-Luciferin, Luciferase) was added to each sample and the plate was incubated at room temperature for 20 min, protected from light. Luminescence intensity was detected by Spark 10 M multimode microplate reader (TECAN), and ATP concentration was calculated according to the manufacturer's instruction.

## Protein extraction and immunoblotting analysis

Total protein was isolated from cells using RIPA buffer containing 25 mM Tris-HCl (pH 8.0), 150 mM NaCl, 1 mM EDTA, 0.5% NP-40, 0.5% sodium deoxycholate, and 0.1% SDS. Protein concentrations were determined using Pierce BCA Protein Assay Reagent. Proteins were separated by SDS-PAGE, transferred to a polyvinylidene fluoride membrane (Millipore Corporation), blocked in 5% fat-free milk for 1 h at RT, and then incubated with primary antibodies in 5% milk overnight at 4 °C. The membrane was then incubated with secondary antibody for 1 h at RT. Antibodies used for western blot analysis were listed in Supplementary Table 1. Primary antibodies and dilutions were used as follows: OPA1 (BD Biosciences, 612606; 1:1000), C18orf19 (FAM210A, Invitrogen, PA5-53146; 1:1000), UCP1 (Abcam, ab10983; 1:200), β-Tubulin (Sigma, T2200; 1:1000), GAPDH (Santa Cruz Biotechnology, sc-32233; 1:2000), Mitofusin 2 (MFN2, Cell Signaling, 9482; 1:1000), OxPhos (Invitrogen, 45-8099; 1:2000), Mitofilin (MIC60, Invitrogen, 45-6400; 1:1000), CHCHD3 (MIC19, Invitrogen, MA5-26597; 1:1000), c-Myc (Santa Cruz Biotechnology, sc-40; 1:1000), YME1L (Proteintech, 11510-1-AP; 1:1000), OMA1 (Proteintech, 17116-1-AP; 1:1000), FLAG (Sigma, F1804; 1:2000), AFG3L2 (Proteintech, 14631-1-AP; 1:1000), LETM1 (Proteintech, 1602-1-AP; 1:1000); GLUT4 (Proteintech, 66846-1-Ig; 1:1000), ATP6 (Proteintech, 55313-1-AP; 1:1000), MTCO2 (Proteintech, 55070-1-AP; 1:1000), CYTB (Proteintech, 55090-1-AP; 1:1000), ND1 (Proteintech, 19703-1-AP; 1:1000). Immunodetection was performed using enhanced chemiluminescence western blotting

substrate (Santa Cruz Biotechnology, cat#sc-2048) and detected with a FluorChem R System (Proteinsimple). The results shown in the figures are representative results from at least three independent experiments.

## Co-immunoprecipitation assay

Total protein was extracted from *Fam210a-Flag* overexpressed HEK293 cells as previously described[77]. The lysate was pre-cleared with protein A/G agarose at 4 °C for 2 h. Then 4 μg of primary antibody anti-Flag was added into cell lysate containing 500 mg total protein and rotating at 4 °C overnight. Protein A/G agarose beads were added into cell lysate and rotated for 2 h at 4 °C. The samples were washed extensively six times and subjected to immunoblotting analysis. Primary antibodies and dilutions were used as follows: FLAG (Sigma, F1804; 1:500), YME1L (Proteintech, 11510-1-AP; 1:300).

## Cell-free protein interaction

FAM210AΔMTS, YME1LΔMTS, OMA1ΔMTS, and OPA1ΔC were expressed in the cell-free protein expression system (NEBExpress cell-free *E. coli* protein synthesis system). The coding sequence of FAM210AΔMTS, YME1L ΔMTS, OMA1ΔMTS, and OPA1ΔC were cloned into NEBExpress control plasmid for protein synthesis. In brief, the coding sequence of mouse FAM210AΔMTS (96−273 amino acids (aa)) was cloned from pCMV6-Fam210a-Myc-DDK; the coding sequence of OPA1ΔC (88−406) with a C-terminal Myc tag was cloned from OPA1-isoform7 (Addgene, plasmid #70179); the coding sequence of YME1LΔMTS (157−716 aa) and OMA1ΔMTS (140−521 aa) was cloned from cDNA of mouse BAT. The proteins were synthesized with 125−500 ng plasmids in a 25-μl reaction system at 37 °C for 4 h with vigorous shaking according to the manufacturer's protocol. Synthesized proteins were mixed (3 μl for each) and incubated in 40 μl reaction buffer[66] (10 mM Tris/HCl (pH 7.4), 50 mM NaCl, 4 mM ATP (pH 7), 8 mM Mg(OAc)$_2$, 50 μM Zn(OAc)$_2$, 2 mM DTT) at 37 °C for the indicated times. Samples were separated by SDS-PAGE and detected by immunoblotting.

## Total RNA extraction and real-time PCR

Total RNA was extracted from tissues using TRIzol reagent (Sigma-Aldrich, cat#T9424) according to the manufacturer's instructions. RNA was treated with RNase-free DNase I to remove contaminating genomic DNA. The purity and concentration of the total RNA were determined by a spectrophotometer Nanodrop 2000c (Thermo Fisher). One microgram of total RNA was reverse-transcribed using random primers with M-MLV reverse transcriptase (Invitrogen, cat#28025021). Real-time PCR was carried out in a Roche Light Cycler 480 PCR System with SYBR Green Master Mix, and gene-specific primers were listed in Supplementary Table 2. The 2$^{-ΔΔCt}$ method[78] was used to analyze the relative changes in each gene's expression normalized against β-actin expression.

## Sample preparation for mitochondria proteomics

Mitochondrial pellets frozen at −80 °C were lysed in PTS buffer (12 mM SLS/12 mM SDC in 100 mM Tris-HCl pH 8.5) and sonicated in 15 W for a total of 1 min with 10 s intervals and 10 s cooling on an ice bath. The mixture was cleared at 16,000 × *g* for 20 min at 4 °C, and the supernatant was transferred into a fresh Eppendorf tube. Protein concentration was measured by BCA assay. Proteins were reduced and alkylated with 10 mM Tris-2-carboxyethyl phosphine and 40 mM 2-chloroacetamide in 100 mM Tris-HCl pH 8.5, with boiling the samples at 95 °C for 5 min. Samples were diluted with 5X trimethylammonium bicarbonate (50 mM). The proteins were digested with lys-C at a 1:100 by mass (enzyme: substrate) for 3 h at 37 °C and then further digested with proteomics grade trypsin to a final 1:100 (enzyme: substrate) ratio overnight. On the next day, samples were acidified with 10% trifluoroacetic acid and desalted with 50 mg Sep-Pak C18 columns

(Waters). A portion of the eluted peptides was quantified with peptide BCA assay, and 2 μg of aliquot peptides were dried to completion in a vacuum concentrator, and samples were ready for Liquid chromatography-mass spectrometry (LCMS) analysis.

## Liquid chromatography (LC)- mass spectrometry (MS) analysis

Peptides were dissolved in 16 μl of 3% acetonitrile /0.3% formic acid, and 4 μl of each sample was injected into a linear ion trap orbitrap mass spectrometer (Thermo Fisher, LTQ-Orbitrap Velos) coupled to an EASY-nLC 1000 HPLC system (Thermo Fisher Scientific) via a nanoelectrospray source was operated in the data-dependent mode for LC-MS analysis of peptides. The electrospray emitter tip was pulled using a laser puller (Sutter Instrument Peptides, Model P2000). Peptides were separated using a C18 capillary column (75 μm inner diameter, 45 cm length) packed in-house with C18 resin (2.2 μm bead size, 100 Å pore size; Bischoff). The mobile phase buffers consisted of buffer A (0.1% formic acid in MS-grade ultra-pure water) and buffer B (0.1% formic acid in 80% acetonitrile). Samples were analyzed using a 90-min gradient from 3% to 30% B, followed by an increase to 50% and 95% B, and then a return to 6% B at a flow rate of 250 nl/min. The mass spectrometer was operated in the data-dependent mode in which a full MS scan from m/z 350–1500 was followed by MS/MS scans of the 10 most intense ions. Ions with a charge state of +1 were excluded, and the mass exclusion time was 60 s.

## Data analysis of proteomic raw files

The raw files were searched against the Mus musculus database with no redundant entries (Uniprot FASTA file released November 2016) using the Andromeda search engine built into MaxQuant[79] (version 1.6.10.43). Andromeda is a probabilistic search engine that uses the LFQ method to identify peptides and proteins with an FDR of less than 1%. MaxQuant generated a reverse decoy database based on the provided forward database, and proteins matching the reverse decoy database or the common laboratory contaminant database in MaxQuant were discarded. The MaxLFQ algorithm was enabled for label-free proteome quantitation, and this feature is a peptide intensity-based algorithm. The Match between runs feature was also enabled to match high-resolution MS1 features between runs, and the retention time window was set to 1.0 min. Peptide precursor mass tolerance for the main search was 6 ppm, and MS/MS tolerance was set to 0.6 Da. Searches were performed with complete tryptic digestion, and peptides were allowed a maximum of two missed cleavages. Search criteria included a static modification of cysteine residues of +57.0214 Da to account for alkylation and a variable modification of +15.9949 Da for potential oxidation of methionine residues.

## Gene ontology analysis

DAVID tool[80] was used for the Gene Ontology (GO) analysis. The $P$ values or adjusted $P$ were reported with the GO terms. The $P$ value was calculated by Fisher's Exact test. Benjamini–Hochberg method was used to adjust the $P$ value. The data was visualized by GraphPad Prism 8.0 (GraphPad Software) and RStudio software as previously described[81].

## Statistical analysis

Statistical analysis and graphing were performed using GraphPad Prism 8.0 (GraphPad Software). Immunoblots were quantified by ImageJ software. The statistical details of the experiments are indicated in the figure legends. Heatmaps were visualized using Heatmapper, average linkage was used for clustering, and Pearson correlation was used for distance measurement. All experimental data are represented as mean ± standard error of the mean (s.e.m.).

Comparisons with $P$ value < 0.05 were considered statistically significant, *$p$ < 0.05, **$p$ < 0.01, ***$p$ < 0.001, ****$p$ < 0.0001.

## Reporting summary

Further information on research design is available in the Nature Portfolio Reporting Summary linked to this article.

## Data availability

The mitochondrial proteomic data generated in this study have been deposited to the ProteomeXchange Consortium via the PRIDE[82] partner repository with the dataset identifier PXD044866. All other data needed to reproduce the work present in this study are available in the manuscript and supplementary information. Source data are provided with this paper.

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

## Acknowledgements

We thank Dr. Shawn S. Donkin, Dr. Kolapo M. Ajuwon, and Linda M. Beckett for assistance on fatty acid oxidation assay, Dr. Heng-wei Cheng for providing the FLIR T440 camera for infrared thermography, Dr. Kolapo M. Ajuwon for assistance on indirect calorimetry study, Dr. Daisuke Kihara for protein structural analysis, Laurie M. Mueller and Robert L. Seiler for assistance on transmission electron microscopy, Donghan Ma for assistance on single-molecule localization microscopy, Meijin Pan and Dr. Di Huang for help with data analysis, and Jun Wu for mouse colony maintenance. This research was supported by NIH-R01CA212609 (S.K.) and NIH-R01DK136722 (F.Y.).

## Author contributions

F.Y., S.K. and J.Q. conceived the project and designed the experiments. J.Q., F.Y., J.C., L.Z., K.K., M.S., N.L. and H.X. performed in vivo experiments and cell-based assays. P.Z., W.T., J.Q. and F.Y. performed mitochondrial proteomic analysis. J.C. and J.Q. performed transmission electron microscopy analysis. F.X. and F.H. performed single-molecule localization microscopy. J.Q., F.Y., J.C., L.Z., K.K., M.S. and N.L., analyzed the data. J.Q., F.Y. and S.K. wrote the manuscript and integrated comments from the other authors.

## Competing interests

The authors declare no competing interests.
