## [Peer Review File · Nature Communications]

FAM210A is essential for cold-induced mitochondrial remodeling in brown adipocytesEditorial Note: This manuscript has been previously reviewed at another journal that is not operating a transparent peer review scheme. This document only contains reviewer comments and rebuttal letters for versions considered at *Nature Communications*.

REVIEWER COMMENTS

Reviewer #1 (Remarks to the Author):

The authors provided new data that addressed the reviewer's comments. No additional comments from this reviewer.

Reviewer #3 (Remarks to the Author):

The authors performed a large number of new experiments to address the previous comments. However, the new data obtained is not unequivocally supporting some of the main conclusions:

1) Authors conclude that FAM210A does not regulate mitochondrial DNA translation in BAT based on:

1a) The absence of immunoprecipitation between FAM210A and translation factors, data not shown and unclear if IPs were performed in cold exposed mice.

1b) The absence of a decrease in mitochondrial protein content and mass in the inducible FAM210A KO mice.

This evidence is not sufficient to discard that the phenotype observed in the constitutive FAM210A KO is not mainly caused by a decrease in the ability to upregulate mitochondrial translation and biogenesis during BAT maturation and expansion, processes that are blocked by thermoneutrality. Indeed, the fact that the inducible KO does not cause a decrease in mitochondrial protein content, nor ATP or membrane potential already demonstrates that loss of FAM210A and the concomitant change in OPA1 processing and

cristae structure are not sufficient to decrease mitochondrial oxidative function or PGC-1 α induced mitochondrial biogenesis.

2) The fact that the BAT mitochondria from the inducible KO show no changes in membrane potential and ATP content after cold exposure supports that FAM210B is dispensable for mitochondrial oxidative function in cold exposed adult BAT. Furthermore, no fatty acid oxidation, energy expenditure, mitochondrial respiration and uncoupling measurements are shown in the inducible KO. The main data related to thermogenic function shown in the inducible KO are an increase in lipid content, changes in mitochondrial cristae structure and OPA1 processing. Given that high fat diet feeding and lipid accumulation in BAT can be associated with higher BAT thermogenic activity (Feldmann et al Cell Metabolism 2009) and that OPA1 processing induced by OMA1 promotes thermogenesis (Quiros et al), it is still a possibility that the inducible KO even showed higher thermogenic activity in BAT.

3) Related to point number 2, the fact that the YME1L-mediated OPA-1 processing is unchanged in the inducible KO supports that YME1L is not activated during cold exposure. If cold exposure activates YME1L, one should see a decrease in band d in control mice, with this decrease being absent in the FAM210B inducible KO. This result challenges one of the main conclusions of the authors: that YME1L proteolytic function is activated by cold exposure to degrade OMA1.

4) The in vitro assays are a great effort, but could the authors comment on why 12 hours are needed for FAM210A to activate YME1L? In addition, how can this model and kinetics be compatible with the fact that the maximal fold increase in OPA1 processing and OMA1 activation in BAT in vivo (after 6 hours cold exposure) coincides with the highest fold increase in FAM210A expression? According to the authors model, one would expect no induction of FAM210A expression during the first 6 hours of cold exposure, allowing OMA1-dependent thermogenesis to proceed. This would be then followed have a large fold increase in FAM210A expression (at 24 hours), when more BAT is being recruited to adapt to long term exposure to cold.

5) The new measurements of Figure 4J,K are confusing. It is stated that these are respiration

measurements in primary brown adipocytes after 6 hours of stimulation with a beta-3 agonist, which should uncouple mitochondria (at least 80%-90%). However, the drop in respiration induced by oligomycin, as well as the increase after FCCP, show a large proportion of coupled respiration, as observed in primary brown adipocytes that were not activated. Rather than decreased thermogenic capacity, these data might be mostly reflecting the decrease in mitochondrial mass and general oxidative function observed in the constitutive KO.

Reviewer #4 (Remarks to the Author):

I appreciate the efforts by the authors to address the concerns raised, and believe the manuscript is now suitable for publication.

Specifically, the authors clarified technical issues associated with tamoxifen injections, added details about the proteome analysis, and increased “n” where requested. ANCOVA analysis was incorporated for the energy expenditure data, and the Seahorse experiments were now performed in primary brown adipocytes. Importantly, the authors performed new experiments to offer additional mechanistic insight regarding FAM210A regulation of YME1L and how that may affect OPA1 processing and cristae remodeling during cold exposure. Although the cell-free system utilized in this revision has limitations that were acknowledged by the authors, the data strengthens the overall conclusions and provides novel information about a sophisticated system regulating cristae remodeling in brown adipocytes to support thermogenesis.

Reviewer #3 (Remarks to the Author):

The authors performed a large number of new experiments to address the previous comments. However, the new data obtained is not unequivocally supporting some of the main conclusions:

We thank the reviewer for appreciating our substantial efforts in revising the manuscript and will try to further convince the reviewer with the additional data described below.

Supplementary Figure 5, new panels (c, d, e, f, g, h) to demonstrate that the inducible KO mice exhibit defects in FAO, thermogenesis, energy expenditure and sympathetic responses upon cold exposure.

New Supplementary Figure 7 to demonstrate that mitochondrial translation was not affected in the inducible KO mice after both short-term (6 h) and long-term (3 day and 7 day) cold exposure.

New texts were also added to the manuscript to describe these new results as well as to outline limitations in response to other points made by the reviewer. We hope the following point-to-point responses (in blue font) will address the reviewer's concerns.

1) Authors conclude that FAM210A does not regulate mitochondrial DNA translation in BAT based on:

1a) The absence of immunoprecipitation between FAM210A and translation factors, data not shown and unclear if IPs were performed in cold exposed mice.

1b) The absence of a decrease in mitochondrial protein content and mass in the inducible FAM210A KO mice.

This evidence is not sufficient to discard that the phenotype observed in the constitutive FAM210A KO is not mainly caused by a decrease in the ability to upregulate mitochondrial translation and biogenesis during BAT maturation and expansion, processes that are blocked by thermoneutrality. Indeed, the fact that the inducible KO does not cause a decrease in mitochondrial protein content, nor ATP or membrane potential already demonstrates that loss of FAM210A and the concomitant change in OPA1 processing and cristae structure are not sufficient to decrease mitochondrial oxidative function or PGC-1alpha induced mitochondrial biogenesis.

We would like to clarify that our intent was not to discard the alternative hypothesis that FAM210A regulates mitochondrial translation, as that hypothesis may be compatible with our hypothesis (that FAM210A regulates OPA1 cleavage) and they don't need to be mutually

exclusive. We presented the evidence (1a and 1b) only to address the reviewer's previous comment on the potential role of FAM210A in mitochondrial translation, as reported in the cardiac muscle ¹. We used the evidence to show that mitochondrial translation does not seem to be a viable avenue to pursue in the brown adipocytes. Now that the reviewer insisted this is a key issue, we performed additional experiments to satisfy the reviewer. Specifically, we conducted cold exposure experiment on the inducible KO mice and collected BAT samples at 6 h, 3 days, and 7 days for western blot analysis of 5 mitochondrial-encoded proteins CI-ND1, CIII-CYB, CIV-MTCO1, CIV-MTCO2, and CV-ATP6 (together with several nuclear encoded proteins). These proteins were selected based on the previous study suggesting that FAM210A regulates translation of mitochondrial-encoded proteins ¹. The results from three biological replicates showed that the expression of these proteins was comparable between control and iKO BAT at all three time points examined (Supplementary Fig. 7). These new data together with our previous evidence (listed 1a and 1b) strongly justify our rationale of not focusing on the role of FAM210A in regulating mitochondrial translation in BAT. We speculate that FAM210A may play cellular context-dependent roles in different cells: while it mediates mitochondrial protein translation in cardiac myocytes that requires substantial ATP production to power heart beating without needing significant cristae remodeling, it mediates OPA1 cleavage to facilitate cristae remodeling in brown adipocytes in response to cold-induced thermogenesis that uncouples ATP production. We included the results in the revised ms.

We addressed in the comment #2 below our response to the second part of the comment "*Indeed, the fact that ... demonstrate that loss of FAM210A and the concomitant change in OPA1 processing and cristae structure are not sufficient to decrease mitochondrial oxidative function or PGC-1alpha induced mitochondrial biogenesis*".

2) The fact that the BAT mitochondria from the inducible KO show no changes in membrane potential and ATP content after cold exposure supports that FAM210B is dispensable for mitochondrial oxidative function in cold exposed adult BAT. Furthermore, no fatty acid oxidation, energy expenditure, mitochondrial respiration and uncoupling measurements are shown in the inducible KO. The main data related to thermogenic function shown in the inducible KO are an increase in lipid content, changes in mitochondrial cristae structure and OPA1 processing. Given that high fat diet feeding and lipid accumulation in BAT can be associated with higher BAT thermogenic activity (Feldmann et al Cell Metabolism 2009) and that OPA1 processing induced by OMA1 promotes thermogenesis (Quiros et al), it is still a possibility that the inducible KO even showed higher thermogenic activity in BAT.

We appreciate the reviewer's comment and agree that additional analysis of the inducible KO mice would substantiate the conclusion. As suggested, we conducted cold exposure experiment with inducible *adipoq^{CreER}-Fam210a^{fllox/fllox}* mice at 5 days after tamoxifen administration. We measured the BAT temperature, energy expenditure, and fatty acid oxidation (FAO) activity in

the iKO mice before and after cold exposure (Supplementary Fig. 5c-h). The results showed that in the absence of cold exposure, no significant changes in energy expenditure, β 3-adrenergic response, and BAT temperature were observed between the iKO and control mice (Supplementary Fig. 5d, e, g). At 6 h post cold exposure, no difference in BAT temperature was observed between the iKO and control mice (Supplementary Fig. 5d). In contrast, at 7 days post cold exposure, the iKO mice exhibited decreased FAO activity, BAT temperature, and β 3-adrenergic response (Supplementary Fig. 5 c, d, f, h), indicating that inducible *Fam210a* KO leads to mitochondrial dysfunction and impairs the thermogenic activity in BAT. These new data, in addition to the previous data from the constitutive and inducible KO mice, demonstrate the indispensable role of FAM210A in cold-induced FAO and thermogenesis in BAT. We have added the new data and results described above in the revised manuscript.

3) Related to point number 2, the fact that the YME1L-mediated OPA-1 processing is unchanged in the inducible KO supports that YME1L is not activated during cold exposure. If cold exposure activates YME1L, one should see a decrease in band d in control mice, with this decrease being absent in the FAM210B inducible KO. This result challenges one of the main conclusions of the authors: that YME1L proteolytic function is activated by cold exposure to degrade OMA1.

We appreciate this thoughtful comment and understand why the reviewer speculated that the OPA1-d band should decrease in the KO. To address this, we quantified the intensities of OPA1-d bands from three biological replicates and indeed found a significant decrease of the level of OPA1-d in iKO BAT upon cold treatment (RL Fig. 1). However, we are cautious about the biological implication of this change and do not plan to include the data in the revised ms. The reason is at least two folded: First, OPA1-d is cleaved from OPA1-a, which contains both S1 (OMA1) and S2 (YME1L cleavage) sites. S1 cleavage generates OPA1-c and S2 cleavage generates OPA1-d. As we don't know the kinetics and interaction of the two cleavage sites (for example if cleavage of S1 affects S2 cleavage), the use of OPA1-d as a surrogate indicator of YME1L activity may be compromised. Second, as the OPA1-c also contain the S2 site (YME1L site), it is possible that it can further be cleaved to OPA1-d. Therefore, the final level of OPA1-d may be dependent on the relative activity of OMA1 and YME1L. In contrast, OPA1-b only has S1 site (OMA1 cleavage site) and OPA1-e can only be produced by OMA1 cleavage OPA1-b, the ratio of OPA1-e/b can be used as a straightforward indicator of OMA1 activity.

Response Letter Figure 1. Relative intensities of OPA1-d bands in control (normalized to 1) and *Fam210a* KO BAT.

4) The *in vitro* assays are a great effort, but could the authors comment on why 12 hours are needed for FAM210A to activate YME1L? In addition, how can this model and kinetics be compatible with the fact that the maximal fold increase in OPA1 processing and OMA1 activation in BAT *in vivo* (after 6 hours cold exposure) coincides with the highest fold increase in FAM210A expression? According to the authors model, one would expect no induction of FAM210A expression during the first 6 hours of cold exposure, allowing OMA1-dependent thermogenesis to proceed. This would be then followed have a large fold increase in FAM210A expression (at 24 hours), when more BAT is being recruited to adapt to long term exposure to cold.

We thank the reviewer for appreciating our *in vitro* assays to directly examine how FAM210a affect OPA1 cleavage. We don't know why 12 h is needed for the effect to manifest but speculate it is due to technical limitations. The cell-free protein interaction system represents a reduced system missing many other components present *in vivo*, as such the reaction condition is far from optimal when compared to the *in vivo* system. Even though in a cell-free system the two proteins can direct interact, without cofactors, adaptors, and regulatory proteins that facilitate conformational changes and protein interactions, the reaction is expected to be inefficient.

For the second part of this comment, we believe that the patterns of FAM210A expression and the OMA1-mediated OPA1 cleavage are compatible with our model. We proposed that upon cold exposure FAM210A functions as a molecular brake by modulating YME1L to prevent the OMA1-mediated over-cleavage of OPA1 in BAT. In supporting this, loss of FAM210A leads to abnormally elevated OPA1 processing at 6 h (Figure 7i and j), indicating that an increased expression of FAM210A is necessary for maintaining OMA1 activity at a proper level at the early stage of cold-induced mitochondrial remodeling. Moreover, after the maximal cleavage of OPA1 at 24 h, the robust increase of FAM210A expression serves to further suppress OMA1 activity, stopping cleavage of OPA1 to enable the restoration of the L-OPA1 at 3 and 7 days as cristae remodeling completes. As the kinetics of L-OPA1 and S-OPA1 and its regulatory proteins remain to be determined, we speculate that the proper suppression of OMA1 activity by FAM210A-YME1L axis is a protective strategy for stepwise OPA1 cleavage to warrant the proper kinetics of OPA1 dynamics.

In addition, it has been well known that the dynamic balance between the L-OPA1 (facilitates fusion and cristae formation) and the S-OPA1 (facilitates fission) is crucial for mitochondrial remodeling under physiological conditions. To facilitate mitochondrial remodeling and augment thermogenic function, acute cold exposure induces rapid mitochondrial fragmentation in BAT⁵. Given that OMA1 deletion decreases the thermogenic activity of BAT, OMA1-mediated cleavage of L-OPA1 to S-OPA1 is required for cold-induced mitochondrial remodeling⁶. On the other hand, abnormal accumulation of S-OPA1 due to prolonged activation of YME1L and/or OMA1 causes mitochondrial damage and cell death⁷. Thus, either too little or too much cleavage

of OPA1 is detrimental, and the cleavage of L-OPA1 must be tightly controlled at a proper to maintain the dynamic balance of L-OPA1 and S-OPA1. Our present study established a new regulatory pathway involving FAM210A-YME1L/OMA1, as a molecular brake, to fine-tune the OPA1 cleavage during cold-induced mitochondrial remodeling in BAT. We have added the above discussion in the revised manuscript.

5) The new measurements of Figure 4J, K are confusing. It is stated that these are respiration measurements in primary brown adipocytes after 6 hours of stimulation with a beta-3 agonist, which should uncouple mitochondria (at least 80%-90%). However, the drop in respiration induced by oligomycin, as well as the increase after FCCP, show a large proportion of coupled respiration, as observed in primary brown adipocytes that were not activated. Rather than decreased thermogenic capacity, these data might be mostly reflecting the decrease in mitochondrial mass and general oxidative function observed in the constitutive KO.

The higher level of coupled respiration in cultured brown adipocytes relative to brown adipose tissue may reflect differences between *in vitro* and *in vivo* systems. First, cultured adipocytes are unable to be fully matured *in vitro* (one example is cultured white adipocytes typically contain multiple lipid droplets as opposed to unilocular morphology of white adipocytes in tissue). Second, the *in vitro* and *in vivo* conditions differ in sympathetic innervation, nutrient sources, mechanical forces, inter-cellular signals, etc. Consistent with these notions, it has been reported that the mitochondria isolated from cultured brown adipocytes are distinct from those in BAT tissue^{8,9}. Specifically, the mitochondria of cultured brown adipocytes cannot be uncoupled as much as those of brown adipocytes *in vivo*. Our Seahorse data are consistent with other studies involving cultured primary beige/brown adipocytes and mitochondria isolated from BAT¹⁰⁻¹² (RL Fig. 2a-c), where most of the data exhibit similar oligomycin-induced drop. Moreover, the description in our manuscript (“reduced UCP1-dependent and maximal respiration”) is based on the quantification using a standard protocol provided by the manufactory (Agilent). We hope the clarification will satisfy the reviewer.

Response Letter Figure 2. Mitochondria of thermogenic adipocytes respond to oligomycin in other studies. **a** *In vitro* cultured differentiated beige adipocytes exhibit oligomycin-induced drop¹⁰. **b** *In vitro* cultured differentiated brown adipocytes exhibit oligomycin-induced drop¹². **c** Mitochondria directly isolated from BAT show oligomycin-induced drop¹¹.

References

1. Wu, J. *et al.* MicroRNA-574 regulates FAM210A expression and influences pathological cardiac remodeling. *EMBO Mol. Med.* **13**, e12710 (2021).
2. Chien, P., Perchuk, B. S., Laub, M. T., Sauer, R. T. & Baker, T. A. Direct and adaptor-mediated substrate recognition by an essential AAA+ protease. *Proc. Natl. Acad. Sci.* **104**, 6590–6595 (2007).
3. Dougan, D. A., Mogk, A., Zeth, K., Turgay, K. & Bukau, B. AAA+ proteins and substrate recognition, it all depends on their partner in crime. *FEBS Lett.* **529**, 6–10 (2002).
4. Graef, M., Seewald, G. & Langer, T. Substrate recognition by AAA+ ATPases: distinct substrate binding modes in ATP-dependent protease Yme1 of the mitochondrial intermembrane space. *Mol. Cell. Biol.* **27**, 2476–2485 (2007).
5. Wikstrom, J. D. *et al.* Hormone-induced mitochondrial fission is utilized by brown adipocytes as an amplification pathway for energy expenditure. *EMBO J.* **33**, 418–436 (2014).
6. Quirós, P. M. *et al.* Loss of mitochondrial protease OMA1 alters processing of the GTPase OPA1 and causes obesity and defective thermogenesis in mice. *EMBO J.* **31**, 2117–2133 (2012).
7. MacVicar, T. & Langer, T. OPA1 processing in cell death and disease—the long and short of it. *J. Cell Sci.* **129**, 2297–2306 (2016).
8. Dyer, R. F. Morphological features of brown adipose cell maturation in vivo and in vitro. *Am. J. Anat.* **123**, 255–281 (1968).
9. Né Chad, M. Development of brown fat cells in monolayer culture: II. ultrastructural characterization of precursors, differentiating adipocytes and their mitochondria. *Exp. Cell Res.* **149**, 119–127 (1983).
10. Wang, Q. *et al.* Post-translational control of beige fat biogenesis by PRDM16 stabilization. *Nature* **609**, 151–158 (2022).
11. Waldhart, A. N. *et al.* Excess dietary carbohydrate affects mitochondrial integrity as observed in brown adipose tissue. *Cell Rep.* **36**, 109488 (2021).
12. Nguyen, H. P. *et al.* Aifm2, a NADH oxidase, supports robust glycolysis and is required for cold-and diet-induced thermogenesis. *Mol. Cell* **77**, 600–617 (2020).

REVIEWERS' COMMENTS

Reviewer #3 (Remarks to the Author):

I want to congratulate the authors for performing these extensive revisions, particularly phenotyping the FAM210A inducible KO. This new set of data clearly demonstrated that FAM210A is required for mitochondrial remodeling induced by long-term cold exposure. In my opinion, the article could be accepted just by changing the title and some conclusions:

1) The absence of defects in thermogenesis and mitochondrial function in the inducible FAM210AKO after a short-term cold-exposure (3 days) strongly supports that differences in cristae remodeling themselves are not responsible for the phenotype of FAM210A deletion. The reason is that mitochondria cristae remodeling and OPA1 processing can occur within minutes of thermogenic activation and change fat oxidation in minutes. A primary and major action on cristae remodeling is expected to change uncoupled respiration in BAT in minutes after activation, and not exclusively after 7 days of cold exposure. 7 days is when mechanisms regulating mitochondrial biogenesis will play a major role.

2) Multiple studies have now demonstrated OMA1 and other mitochondrial proteases can generate peptides that activate transcriptional programs and stress responses (HRI-DELE1). OMA1 overactivation for instance can activate the mitochondrial ISR to suppress both mitochondrial biogenesis and mitophagy. In this regard, a blockage in both biogenesis and mitophagy can explain the new data absence of changes in protein content 7 days after cold exposure in the inducible KO. The fact that the phenotype in the inducible KO is only observed 7 days after cold exposure, and not before (3 days), supports that differences in stress responses and transcriptional reprogramming are explaining the phenotype. Indeed, in the constitutive KO authors already see and conclude that: "our finding suggests that Fam210aAKO impairs cold-induced mitochondria biogenesis." Therefore, maybe FAM210A restrains OMA1 activity to prevent the mitochondria ISR and allow mitochondrial biogenesis activated by long-term cold exposure to proceed. These two points should be added to data interpretation and the discussion.

3) A suggested title that informs the phenotype better would be:

FAM210A is essential for cold-induced remodeling of BAT mitochondria.

Such a change is essential, as with the new data, authors cannot discard still if cristae structure or regulation of mitochondrial biogenesis is the main action of FAM210A determining BAT mitochondrial function after long-term cold exposure.

4) The response to the respirometry traces is still inadequate. Authors are missing the basic knowledge that UCP1-controlled respiration cannot be coupled and thus cannot respond to oligomycin. I urge the authors to correct this, as it is basic bioenergetics. Published references can be wrong if the paper was not reviewed by researchers experienced in bioenergetics and BAT primary cultures.

REVIEWERS' COMMENTS

Reviewer #3 (Remarks to the Author):

I want to congratulate the authors for performing these extensive revisions, particularly phenotyping the FAM210A inducible KO. This new set of data clearly demonstrated that FAM210A is required for mitochondrial remodeling induced by long-term cold exposure. In my opinion, the article could be accepted just by changing the title and some conclusions:

We thank the reviewer for constructive comments. We changed the title and revised the conclusions as suggested.

1) The absence of defects in thermogenesis and mitochondrial function in the inducible FAM210AKO after a short-term cold-exposure (3 days) strongly supports that differences in cristae remodeling themselves are not responsible for the phenotype of FAM210A deletion. The reason is that mitochondria cristae remodeling and OPA1 processing can occur within minutes of thermogenic activation and change fat oxidation in minutes. A primary and major action on cristae remodeling is expected to change uncoupled respiration in BAT in minutes after activation, and not exclusively after 7 days of cold exposure. 7 days is when mechanisms regulating mitochondrial biogenesis will play a major role.

We appreciate the clarification and revised the conclusions of the related cold-exposure results in the manuscript (Line 275-280).

2) Multiple studies have now demonstrated OMA1 and other mitochondrial proteases can generate peptides that activate transcriptional programs and stress responses (HRI-DELE1). OMA1 overactivation for instance can activate the mitochondrial ISR to suppress both mitochondrial biogenesis and mitophagy. In this regard, a blockage in both biogenesis and mitophagy can explain the new data absence of changes in protein content 7 days after cold exposure in the inducible KO. The fact that the phenotype in the inducible KO is only observed 7 days after cold exposure, and not before (3 days), supports that differences in stress responses and transcriptional reprogramming are explaining the phenotype. Indeed, in the constitutive KO authors already see and conclude that: "our finding suggests that Fam210aAKO impairs cold-induced mitochondria biogenesis." Therefore, maybe FAM210A restrains OMA1 activity to prevent the mitochondria ISR and allow mitochondrial biogenesis activated by long-term cold exposure to proceed. These two points should be added to data interpretation and the discussion.

We thank the reviewer for the insights into potential mechanisms underlying the observed phenotypes and included some discussion to reflect these (Line 441-450).

3) A suggested title that informs the phenotype better would be:

FAM210A is essential for cold-induced remodeling of BAT mitochondria.

Such a change is essential, as with the new data, authors cannot discard still if cristae structure or regulation of mitochondrial biogenesis is the main action of FAM210A determining BAT mitochondrial function after long-term cold exposure.

We thank the reviewer for the suggestion. The title has been as suggested

4) The response to the respirometry traces is still inadequate. Authors are missing the basic knowledge that UCP1-controlled respiration cannot be coupled and thus cannot respond to oligomycin. I urge the authors to correct this, as it is basic bioenergetics. Published references can be wrong if the paper was not reviewed by researchers experienced in bioenergetics and BAT primary cultures.

We revisited our previous responses and saw where the reviewer's comment came from. The reviewer believed that the coupled respiration (which is blocked by Oligomycin) should be less than 10-20% of the maximal respiration (combined coupled and uncoupled) as the uncoupled respiration should account for at least 80-90% of the total respiration in brown adipocytes. In our case the oligomycin-sensitive OCR accounts for ~30% of maximal OCR, this value is consistent with other studies using cultured BAs. We believe that the relative larger coupled respiration is probably due to contamination of non-brown adipocytes in the primary culture. It is very common that SVF contains a significant fraction of fibroblasts that cannot be differentiated into brown adipocytes. We clarified this result in Line 210 – 216 of the revise manuscript. We would like to emphasize that the main point here is that the oligomycin-sensitive coupled respiration is similar in Fam210a KO and WT cells, yet the UCP1-dependent uncoupled respiration is reduced in the KO cells.